# The genomic impact of population connectivity and decline in Africa's elephants

Patrícia Pečnerová [1,2,3] ✉, Yasuko Ishida [4], Genís Garcia-Erill [1,5], Laura D. Bertola [1,6], Cindy G. Santander[1], Xiaodong Liu [1], Anna Brüniche-Olsen [7], Anubhab Khan [1,8], Lauren M. Hennelly[9,10,11], Renzo F. Balboa[1], Long Lin[1], Malthe S. Rasmussen[1], Xi Wang [1], Mikkel Schubert [12], Amal Al-Chaer[1], Sylwia Urbaniak[13], Kan Nobuta[13], Sasha Treadup[13], Karine A. Viaud-Martinez [13], Alida de Flamingh[14], Ripan S. Malhi [14,15,16], Savanah Bird[17], Nelson Ting[17], Mrinalini Watsa [18], Martin N. Tchamba[19], Stéphanie Bourgeois[20], Chris Thouless[21], Iain Douglas-Hamilton[21,27], George Wittemyer [21,22], Peter J. Van Coeverden de Groot[23], Vincent B. Muwanika [24], Charles Masembe [25], Nicholas J. Georgiadis[26], Christina Hvilsom[3,9], Rasmus Heller [1,28], Hans Redlef Siegismund[1,28] & Alfred L. Roca[4,14,28]

African elephants are keystone species facing severe declines due to the ivory trade and habitat loss. To investigate the genomic consequences, we analyze 232 high-coverage genomes from 17 African countries in the first continent-wide genomic analysis treating savanna (*Loxodonta africana*) and forest (*L. cyclotis*) elephants as distinct species. We find a deep divergence between species, with forest elephants showing higher heterozygosity and historically larger effective population sizes, while savanna elephants exhibit greater inbreeding and genetic load. Surprisingly, we detect widespread introgression of trace forest ancestry across savanna populations, suggesting a complex history of hybridization. Within species, historically high mobility promoted genetic connectivity, though we identify signs of human-induced isolation and drift in peripheral populations. Our findings highlight gene flow as a key force in African elephant evolution and underscore the urgency of understanding the impact of accelerating habitat fragmentation in these ecosystem engineers.

Megaherbivores are especially vulnerable to extinction[1,2], and the loss of large land mammals can have cascading effects on entire ecosystems[3]. The largest of them all, elephants, are among the most iconic species threatened by extinction in the Anthropocene[4]. Elephants in Africa are now recognized as two species, the African savanna elephant (*Loxodonta africana*) and the African forest elephant (*L. cyclotis*). Both species are considered threatened due to poaching for ivory[5], habitat loss and fragmentation[6,7], as well as the ensuing human-wildlife conflict[8]. According to the 2020 Red List assessment of

the International Union for Conservation of Nature (IUCN), the first to evaluate African elephants as individual species, the savanna elephant is listed as "Endangered"[9] and the forest elephant is listed as "Critically Endangered"[10].

The question of how many species of elephant are present in Africa has dominated elephant genetics research. In the 1930s, scientific discussion began with arguments in favour of recognizing African elephants as two species due to morphological and other differences[11–13], while arguments against it emphasized evidence of

hybridization[14]. The discussion was revived in 2000 when Groves & Grubb[15] demonstrated that savanna and forest elephants are morphologically distinct based on quantifiable skull measurements, and Roca et al.[16]. found limited interspecies gene flow despite the otherwise long-distance within-species gene flow. The first genomes, two from each species, confirmed that they are two independent evolutionary lineages that diverged 2–5 million years ago, and little if any gene flow was estimated to occur between the two species after ≈600,000 years ago[17].

Despite the deep split between the savanna and forest elephant, hybridization does occur in a number of locations where their habitats overlap[15,16,18–20]. Importantly, genetic analyses have revealed that hybridization is limited and only 6.6% of 2445 geo-referenced samples collected across Africa were of hybrid origin. In fact, hybrids were found only in areas in or near savanna-forest ecotones[21] and 70% of geo-referenced hybrids (117/167) were found only in the main hybridization zone along the Albertine Rift on the border between the Democratic Republic of Congo (DRC) and Uganda[21].

It has been hypothesised that environmental factors might have contributed to hybridization between savanna and forest elephants, and had a general formative effect on elephant evolutionary history[20,22–24]. The climatic changes associated with glacial cycles in the Pleistocene led to habitats expanding and contracting and resulted in rapid environmental shifts[25]. During the colder and drier conditions of the arid phases, previously continuous rainforests retracted and fragmented, leaving forest-adapted species isolated in refugia[26,27]. When the arid phases ended, forests expanded again, and forest elephants may have come into contact with the savanna elephants temporarily occupying the area, enabling hybridization. Conversely, the expansion of forests in humid interglacial phases could have led to a temporal separation of savanna-dwelling species[28–30]. It has been proposed that isolation in refugia might explain the deep subdivision of mitochondrial clades or the high diversity of forest elephants as diverged populations came back into secondary contact[22–24,31].

On a shorter time scale, elephant numbers and range have been mainly shaped by human activity. The combined effects of ivory trade and habitat loss have driven large-scale population declines[9,10]. Poaching has often led to episodes of rapid decline, while continuous erosion of habitat has led to a slower, long-term decline. Killing for ivory has fluctuated over the last centuries with several major peaks. In the 19th to early 20th century, exploitation intensified due to an increased demand for ivory in Europe and North America[32]. In the 1970s and 1980s, the price of ivory rose sixfold within a decade, and armed conflicts related to civil unrest across the continent brought an influx of automatic rifles[33]. The most recent poaching crisis started in 2007 due to high demand for ivory in the rapidly growing economies of Asia[34].

At the same time, human population density increases, land use changes, and infrastructure growth in Africa have been accelerating[35]. Between 1960 and 2024, Africa experienced the largest relative human population growth of all continents, increasing fivefold from 283 million to 1.5 billion people[36]. As a consequence, remaining elephant populations occupy only fragments of their original range, distributed in patches of increasingly isolated habitat. It has been estimated that 62% of the African continent is potentially suitable elephant habitat, of which elephants use 17%[37]. Moreover, the human population in sub-Saharan Africa is projected to further triple by 2100[38–40], and continued reductions in viable habitat are anticipated.

The impact of anthropogenic activities can be clearly seen in diminishing elephant numbers. In a recent analysis of survey data from 475 sites across 37 countries, Edwards et al.[41] modelled elephant density trends over a 53-year period, estimating an average decline of 70% for savanna elephants and 90% for forest elephants. There are no precise estimates of how many elephants lived in Africa historically, but ≈1.3 million elephants were surviving as late as 1979[42,43]. According to the most recent surveys, only ca. 415,428 savanna and forest elephants remain in the wild in surveyed areas, with potentially 117,127 to 135,384 more in areas not systematically surveyed[44]. Because African elephants were assessed as two separate species only as recently as 2021, species-level information is particularly scarce.

Anthropogenic pressure has shaped the elephant population trajectories and influenced elephant populations in different regions of Africa in different ways, creating a mosaic of population histories. While the history of elephant population declines has been relatively well documented, the impact on genomic makeup and how it affects the survival, reproduction, and adaptability of the species has been barely studied. Understanding the genomic consequences of these processes, and how they influence the risk of extinction is one of the most pressing questions in conservation genetics[45]. We are quickly gaining insights into the genomic consequences of population declines in many species of conservation concern[46–48], but also in extinct species, including extinct elephantids like the woolly mammoth[49]. However, we know comparatively little about the genomic status and genetic risk factors faced by African elephants.

To facilitate the transition from genetics to genomics in elephant research, we harness existing curated collections of samples that were obtained in the 1990s[23,50], and from these, we sequence 225 high-quality genomes. This allows us to perform a continent-wide genomic analysis of African elephants, studying the African savanna and forest elephants jointly but as separate species. This offers a finer-scale resolution of population structure and reveals broad-scale introgression of trace amounts of forest ancestry across savanna populations, while also providing genome-wide estimates of heterozygosity, estimates of inbreeding from runs of homozygosity and genetic load as a proxy for fitness effects. Originating from the 1990s, this dataset captures elephant populations approximately one generation before today, predating the most recent poaching crisis of the 21st century. Thus, it both reflects the consequences of human-induced population declines over the last centuries and also serves as a baseline against which any future studies of present-day populations can be compared.

## Results and discussion

We analyzed a dataset of 249 whole genomes (Supplementary Data 1), including 225 newly sequenced genomes and 24 previously published genomes[17,51,52]. The genomes were primarily generated from biopsy tissue and blood samples collected in the 1990s for previous studies[16,23,50,53]. We mapped the sequencing data to two chromosome-level genome assemblies, the African savanna elephant (LoxAfr4[17]) and the Asian elephant (mEleMax1; Vertebrate Genome Project; Bioproject: PRJNA850184). Rigorous quality control was applied to identify genomic regions with variants that could be reliably called in either reference, resulting in 1.2 billion sites available for analyses in the African savanna elephant reference and 1.3 billion sites in the Asian elephant reference.

During initial quality control, 17 samples were identified as either low quality or duplicates (Supplementary Data 2). After removing these samples, the final dataset resulted in 232 elephants sequenced at an average depth of 39× when mapping to the LoxAfr4 reference, comprising 181 African savanna elephants and 51 African forest elephants from 29 locations across 17 countries (Figs. 1 and S1). Out of the 232 genomes, 207 were sequenced to more than 9.5×, and these were included in analyses requiring genotype calls. In addition, we performed imputation of missing genotypes for all 232 genomes. Thus, our analyses were performed on one of two types of datasets—genotype calls of 207 individuals or imputed genotypes for all 232 individuals (Supplementary Data 3). During quality control, we identified eight samples as first-degree relatives of other samples. We included these samples in individual-based analyses, but excluded the lower coverage individual from each pair in analyses based on allele frequencies (Supplementary Data 3).

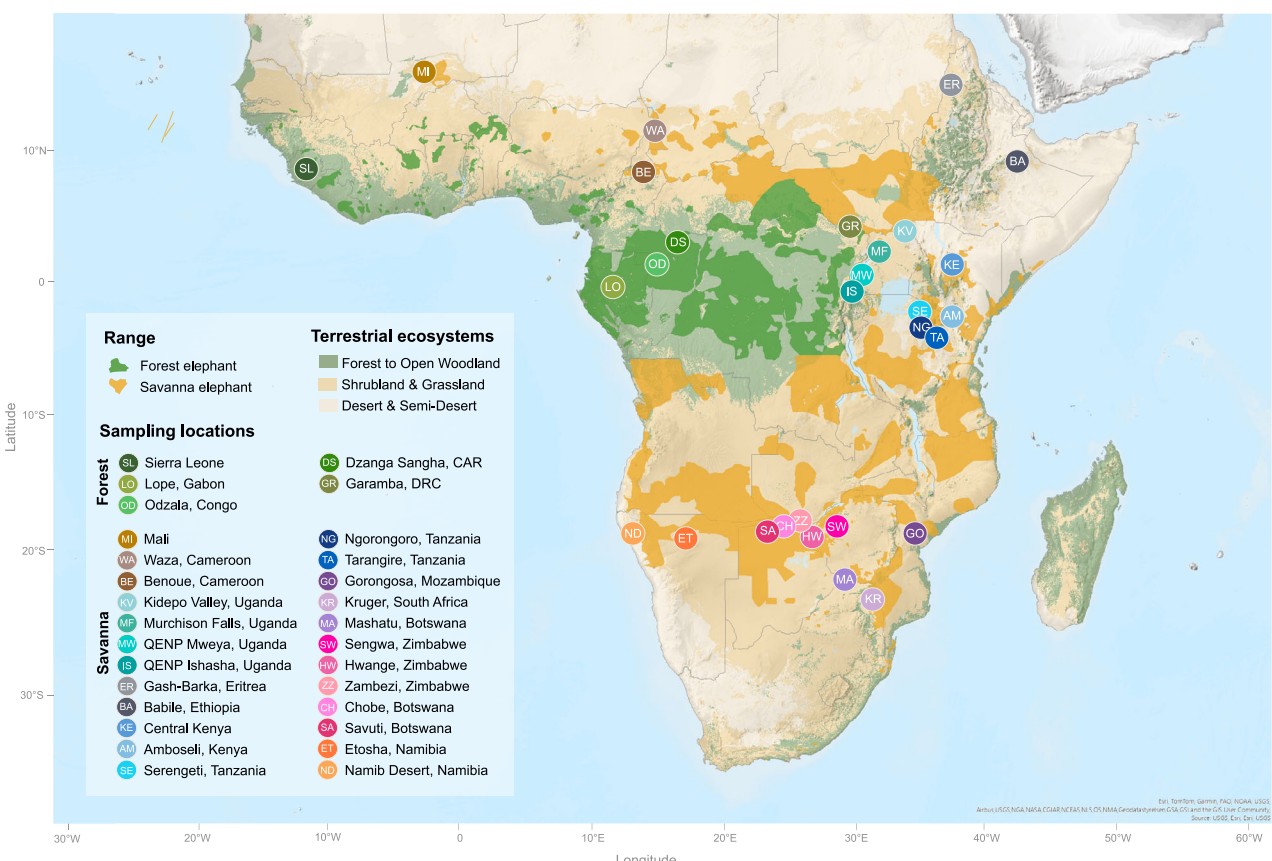

**Fig. 1 | A map of sub-Saharan Africa showing the locations of origin of the samples** analysed in this study plotted against the distribution of terrestrial ecosystems[165] and the current species ranges, including possible ranges, according to the IUCN Red List of Threatened Species[9,10]. An extended map, that includes sampling sizes and locations that were excluded because representative samples failed quality control, is shown in Fig. S1. The map was created with ArcGIS® Online. Basemap and spatial data © Esri and its data providers[166–169]. The map was edited in InkScape v1.2.1.

## Two distinct species, with hybridization

Population structure in African elephants was dominated by the contrast between the deep divergence separating the two species, the savanna and the forest elephant, and the relatively weak structure within each species. Using the imputed dataset of 232 elephants, we found that the main axis of variation (PC1) inferred in a principal component analysis (PCA) performed in PLINK v1.9[54] separated savanna and forest elephants, explaining 86.7% of the variance in the data (Figs. 2a and S2). Further substructure within each of the two species is evident in PC2 and PC3, albeit with much lower proportions of explained variance (Fig. 2a).

In analyses of ancestry proportions performed in ADMIXTURE[55], the savanna and forest elephant formed two distinct clusters at $K = 2$ (Figs. 2b and S3). The analysis converges up to $K = 6$, but evaluation of the model fit in EvalAdmix[56] revealed that the clustering at $K = 6$ did not improve compared to $K = 5$ (Fig. S4), and therefore we show the results at $K = 5$. At $K = 5$, the forest elephants were found to be further split with one partition encompassing forest elephants in localities within the western part of the Congo Basin rainforest (henceforth the central forest), along with the sole West African forest elephant sample (henceforth west forest); and the other partition consisting of Garamba in the eastern DRC (henceforth east forest). At $K = 5$ (Fig. 2b), savanna elephants were also partitioned into southern, eastern and west-central regions (west-central is used to designate Mali and Cameroon localities with savanna elephants). Equivalent clusters were observed at higher PCs of the PCA (Fig. S2). Finally, the $F_{ST}$ between species, calculated as the average pairwise $F_{ST}$ between locations of the two species, was 0.64. By contrast, the within-species $F_{ST}$, calculated as

the average of all pairwise comparisons between locations within each species, was much smaller: 0.067 in savanna elephants and 0.054 in forest elephants (Fig. S5). This aligns with the contrast between the high proportion of variance (86.7% in PC1) in the data explained by the difference between the species and the low proportion of variance (1.4% for PC2; 1.0% for PC3) explained by differences within species (Fig. 2a).

On the other hand, we found indications of gene flow occurring between the savanna and forest elephants at different points in time from the deep past until recently. The most noticeable was the hybridization at the DRC-Uganda border (in the Queen Elizabeth National Park, QENP, and specifically in the Mweya population) and in the Garamba National Park, which was reflected in the gradient of mixed ancestries in PCA and Admixture analyses (Fig. 2).

We investigated gene flow between species using $D$-statistics[57,58] on the dataset mapped to the Asian elephant. Prompted by the observation of forest ancestry in some individuals in Serengeti and Zambezi in the Admixture analyses (Fig. 2 and Supplementary Data 4), we tested whether any savanna elephant population showed an excess of allele sharing with the forest elephant. We placed forest elephants (excluding the admixed Garamba population) as H3 and all combinations of savanna elephants in H1 and H2. All $D$-statistics were summarized by fixing a relatively non-admixed savanna population geographically distant from the forest populations (Kruger) as H1, a relatively non-admixed forest population (Lope) as H3, and averaging each population as H2. Unsurprisingly, we found that the populations in recognized hybrid zones in QENP, Garamba, and west-central Africa[18,20] showed strong signals of allele sharing (Figs. S6–S8),

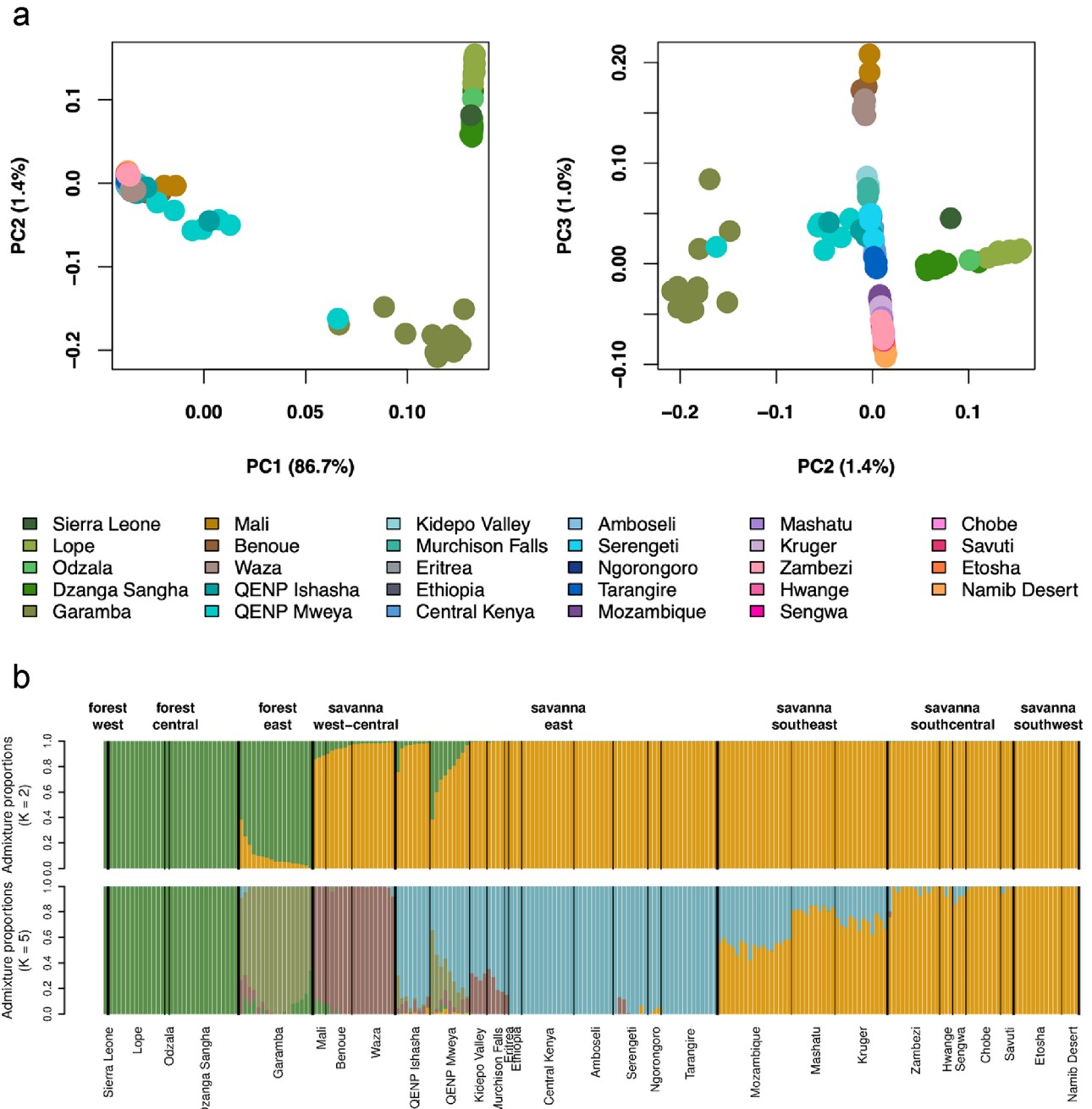

**Fig. 2 | Population structure of both species of African elephants based on the imputed dataset of 232 savanna and forest elephants. a** Principal component analysis reflecting principal components (PCs) 1 to 3. PC 1 reflects the variance between savanna and forest elephants, PC 2 reflects the separation within forest elephants, and PC 3 shows an "isolation-by-distance-like" pattern in savanna elephants. Higher PCs can be found in Fig. S2. **b** Admixture analysis at $K = 2$ and $K = 5$ are shown. Additional Ks are shown in Fig. S3.

consistent with the evidence of recent admixture observed in the PCA and Admixture analyses (Figs. 2, S3, and S9). We corroborated the timing of this recent hybridization using *apoh*[59]. In QENP and Garamba, we found several individuals that had parents with different admixture proportions, indicative of recent hybridization two to six generations ago, while others, as well as elephants in Mali, show hybridization more than six generations ago (Figs. S10 and S11 and Supplementary Data 5).

To infer the proportion of forest ancestry among savanna elephant populations, we used the F4 ratio test in ADMIXTOOLS2[60] in the form of $\alpha =$ f4 (Sierra Leone, Asian; Kruger, target)/f4 (Sierra Leone, Asian; Kruger, Lope and Dzanga Sangha). We also compared this estimate to the proportion of forest ancestry based on Admixture at

$K = 2$ and based on the PC1 (Supplementary Data 4 and Fig. S12), and we found that estimates of all three methods showed consistent results. Our estimates (Fig. 3 and Supplementary Data 4) suggest that the savanna elephants in QENP Mweya carry more than 20% forest ancestry and that the forest elephants in Garamba have more than 10% savanna ancestry. We also estimated ≈15% forest ancestry in savanna elephants in Mali, ≈6% in QENP Ishasha and Benoue, Cameroon, and ≈2% in Waza, Cameroon.

Importantly, we also found that many savanna populations outside of the recognized hybrid locations carry trace-amounts of forest elephant ancestry, suggesting a legacy of gene flow between savanna and forest elephants in the deeper past. The signal of forest ancestry

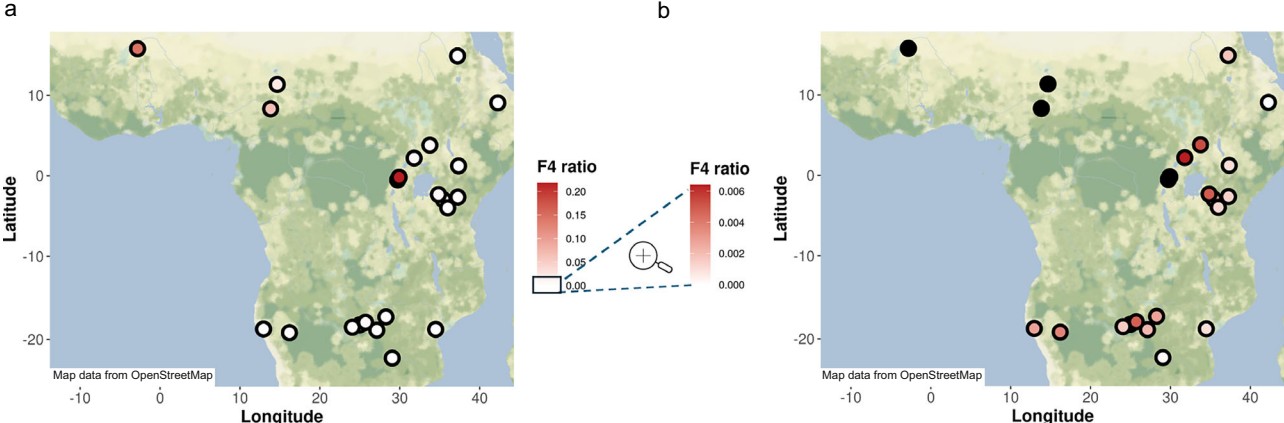

**Fig. 3 | A map summarizing the patterns of forest ancestry among the savanna elephant populations inferred using the F4 ratio test. a** Higher proportions of forest ancestry are evident in west-central Africa and QENP. **b** Color saturation enhances a weaker signal of forest ancestry in populations outside of the recognized hybrid zones. An alternative map, showing D-statistics of patterns of forest allele sharing among the savanna elephant populations is available in Fig. S6. The map was generated in R v4.2.2 using R package ggplot2[170] and the Stamen Terrain basemap accessed via Stadia Maps. Map tiles: © Stadia Maps © Stamen Design © OpenMapTiles © OpenStreetMap contributors. Map data: © OpenStreetMap contributors (ODbL).

was most notable in elephants of northern Uganda, Serengeti and Zambezi, which carry ≈0.5% forest ancestry (Fig. 3b and Supplementary Data 4), consistent with forest ancestry being detectable in some individuals in these populations in the Admixture analyses (Fig. 2 and Supplementary Data 4). While minuscule, the inferred forest elephant ancestry in the genomes of savanna elephants was widespread geographically, and a Mantel test revealed a weak but significant correlation between forest ancestry and geographic distance to the Congo-Guinean forest (Spearman's $\rho = 0.234$, $p = 1e^{-04}$; Fig. S13), indicating that forest elephant-associated ancestry tends to decline with increasing distance from the tropical forest. It remains unclear if the forest elephant ancestry spread through the savanna range by stepwise gene flow from the currently recognized hybrid zones, or if it reflects a different geographic position of the hybrid zones in the past, when the extent of the tropical forest was broader.

The ancient hybridization was previously linked to changes in the extent of tropical forest during glacial cycles and a geographically shifting hybrid zone[26,61,62], consistent with the presence of forest elephant-like haplotypes within populations of savanna elephants[53,63]. Roca et al.[63] also observed that the percentage of forest elephant-like mtDNA in savanna elephants in Tanzania drops with increasing distance from the current expanse of the tropical forest. Alternatively, or in parallel, forest elephant nuclear alleles may be spread among savanna populations from the present-day hybrid zone by gene flow, consistent with observations that elephants can move over long distances and have home ranges of up to 5000 km² [64]. In addition, it is known that some populations, for instance, Serengeti, have a complex population history including recolonizations[65]. It is thought that the Serengeti population was recolonized from the Lambwe Valley in Kenya[66] and our findings support the presence of Kenyan ancestry in Serengeti, as elephants from Serengeti are genetically closer to elephants from Central Kenya ($F_{ST} = 0.009$) and to smaller extent the Amboseli National Park ($F_{ST} = 0.015$) than to the neighbouring Ngorongoro ($F_{ST} = 0.016$) or Tarangire ($F_{ST} = 0.021$; Figs. 4 and S5). Curiously, since Serengeti has more forest elephant alleles compared to the Kenyan populations, the forest elephant ancestry in Serengeti did not originate from the recolonization from Kenya in the 1950s.

We also tested whether any forest elephant population showed an excess of alleles shared with the savanna elephant. We tested all combinations of forest elephants in H1 and H2, and placed savanna elephants in H3 as a relatively non-admixed savanna elephant population (Kruger). As expected, we detected a signal of savanna elephant

ancestry in the hybrid population of Garamba; however, we also detected signals of savanna ancestry in Dzanga Sangha and Odzala (Fig. S14), suggesting that the gene flow is bidirectional even outside of the hybrid zone, although this was not evident in PCA or Admixture analyses (Fig. 2). However, the coarse representation of forest elephants in our dataset, and the lack of savanna elephant mtDNA introgression in forest elephants[23], prevents us from drawing conclusions about the broader patterns of savanna introgression in forest elephants.

A potentially complex pattern of bidirectional gene flow between species was corroborated by admixture graph analyses in qpGraph in ADMIXTOOLS2[60] (Fig. S15) and Treemix[67] (Fig. S16), and this makes inferences regarding the phylogeny of African elephants challenging, as illustrated by the basal placement of the hybrid populations of Garamba and the Queen Elizabeth National Park in the autosomal phylogenetic tree inferred using IQ-TREE[68,69] (Figs. S17–S19), which is most likely driven by the high proportion of mixed ancestry.

Overall, these patterns of introgression between the two African elephant species suggest gene flow occurring at different points in time. Palkopoulou et al.[17]. concluded that species-wide gene flow between savanna and forest elephants ceased at least 600,000 years ago, but their estimate was based on comparing only two forest elephants (from Sierra Leone and the Central African Republic) and two savanna elephants (from Kenya and South Africa). Analyzing a much broader representation of each species, we see evidence of the recent admixture happening within the last two centuries, including recent gene flow along the DRC-Uganda border and in Garamba, and an older gene flow in west-central savanna elephants (Figs. 2 and S10 and Supplementary Data 5). However, there is also evidence of admixture in the deeper past as documented by the presence of forest-like mitochondrial haplotypes and small proportion of forest ancestry among savanna populations across the range.

All populations in the recognized hybrid locations have elevated levels of genome-wide heterozygosity measured as the proportion of heterozygous genotypes (Figs. 5 and S20 and Supplementary Data 6) and lowered inbreeding coefficient estimated from runs of homozygosity (ROH; Fig. 6). Moreover, even in populations with low levels of forest ancestry, for example Serengeti and Zambezi, we observed a slightly elevated heterozygosity ($H_O = 0.87 \times 10^{-3}$ in Serengeti and $H_O = 0.88 \times 10^{-3}$ in Zambezi). However, the high levels of genetic variation should not be seen as justification for, or validation of, anthropogenic hybridization. The consequences of the hybridization related

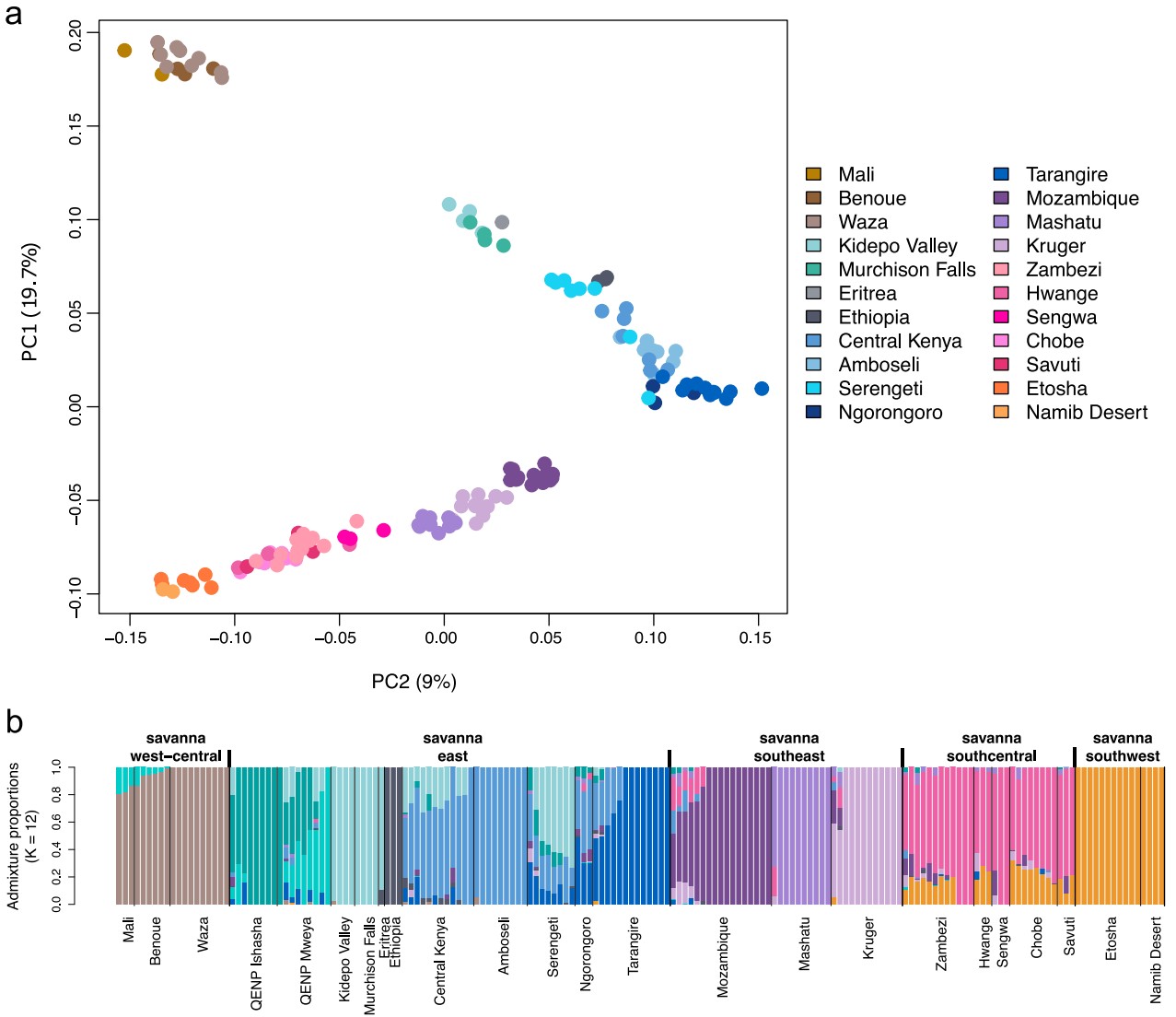

**Fig. 4 | Population structure of savanna elephants based on called genotypes.**
**a** Principal component analysis performed in PLINK without the QENP samples. Higher principal components (PCs) can be found in Fig. S26. Note that PC1 and PC2 are switched compared to Fig. 2 in order to highlight how genetic variation follows the geography of regions and localities. **b** Admixture analysis also demonstrated structuring by region and locality; *K* = 12 is shown. Lower Ks are shown in Fig. S27.

to fitness, health, behaviour, social structure and other aspects of elephant biology remain unknown, and only the first strides in characterizing hybrids phenotypically are being made[70]. While hybridization can be a natural process that has played an important role in the evolutionary history of proboscideans[17,71], including an ancient hybrid origin of the Columbian mammoth[72], elephant history in the last millennia has been drastically shaped by human activities, and separating natural from anthropogenic drivers of hybridization in elephants is challenging. One possible way to shed light on the driving factors is by timing the hybridization. Our results from *apoh* suggest that hybridization occurred two to six generations ago in Uganda and Garamba, and more than six generations ago in Mali, perhaps suggesting a role for human activity. While further genomic analyses are necessary to address this complex issue in depth, as well as assess the role of selection in the introgressed regions, our findings expand our understanding of the genetic makeup of species with a history of hybridization. Understanding the causes and consequences of hybridization is becoming increasingly important as hybridization is a process that might accelerate in a world of human-induced range shifts[73], and addressing "the problem with hybrids"[74] in conservation policy, including in elephants[75], is still a matter of scientific debate.

## A mosaic of population histories

Perhaps even more than between the species, the formative effect of gene flow is obvious within species. As a result of the homogenizing effect of within-species gene flow, we detect only weak population structure, which is in contrast to the more profound differences between the species. In savanna elephants, the population structure appears to be consistent with isolation by distance (Figs. 4, S21, and S22), a pattern also observed in pan-African genomic analyses of lions[76], leopards[77], and Cape buffalo[78]. This pattern is consistent with the long-distance dispersal of these species, in which philopatric females maintain related social groups, while dispersing males move alleles across the landscape, resulting in a gradient of genetic differentiation with distance. Importantly, even the weak structure can be captured thanks to the high resolution of whole genome data (Fig. 4).

In forest elephants, our limited sampling allows only a broad assessment of population structure, especially since two of the five sampled locations are represented by a single individual. Both the PCA (Fig. S23) and Admixture analysis (Figs. S24 and S25) showed a split between the forest elephants in west (Sierra Leone) and central Africa (Lope, Dzanga Sangha, Odzala) and those of east Africa (Garamba), as was previously indicated in a study using 18 microsatellite markers[31].

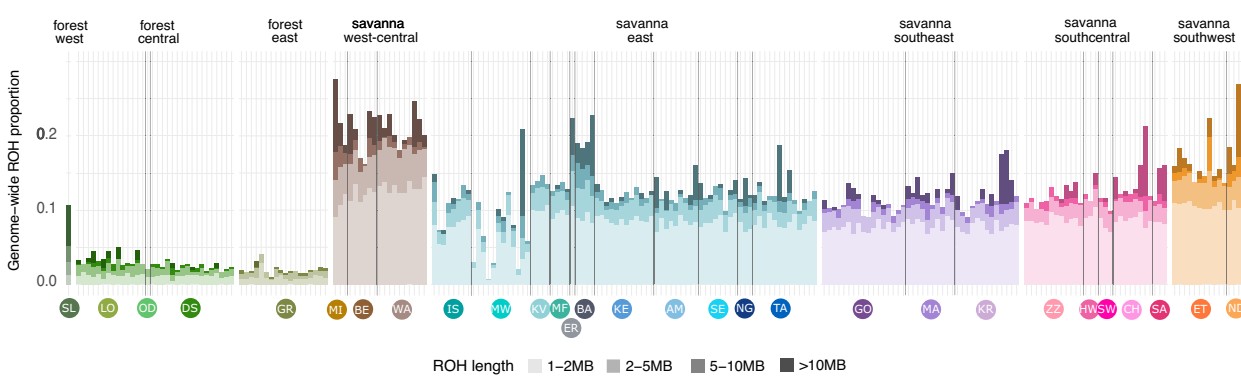

**Fig. 5 | Genome-wide heterozygosity estimates from called genotypes.** Predominantly savanna populations are in orange; predominantly forest populations are in green. Hybrid populations exhibit elevated heterozygosity. Within-species heterozygosity levels are generally consistent, with forest elephants showing higher values than savanna elephants. Isolated populations tend to have lower heterozygosity. Boxplots summarize the distribution of the data using the median (center line) and interquartile range (25th–75th percentiles; the box), with whiskers extending to 1.5× the interquartile range; outliers are shown as individual points.

Individual heterozygosity values, along with population information, which were used for the calculation of per-population heterozygosities are available in Supplementary Data 6. The maps illustrate putative groupings based on the Admixture analyses, though most partitions are shallow, with low $F_{ST}$ values indicating low genetic differentiation. The map was manually generated in InkScape v1.2.1 based on an outline of Africa generated in R v4.5.1 using ggplot2[170], dplyr[171], sf[172,173], and rnaturalearth[174] R packages. Elephant silhouettes have been used from PhyloPic under CC0 1.0 Universal licence.

**Fig. 6 | Runs of homozygosity (ROH) estimated in PLINK.** Savanna elephants have overall higher levels of inbreeding than forest elephants. Populations known to be more isolated have a higher proportion of ROH. On the other hand, savanna

elephants from QENP (IS, MW) have relatively low proportions of ROH due to recent admixture with forest elephants. See Fig. 1 for legend on location codes. A version including individual labels is available in Fig. S30.

However, it is important to note that the clustering of the west African forest elephant from Sierra Leone with central forest elephants is likely driven by the former being a singleton. The weak population structure among central forest elephants is consistent with expected high connectivity between these populations. The Lope, Dzanga-Sangha and Odzala-Kokoua National Parks form part of the largest stronghold for forest elephants[79–81]. In addition, these forest elephants had been relatively preserved from hunting until recent times due to remote, inaccessible habitat and lower density of the human population[82].

Analyzing savanna elephants separately from the forest elephants, we identified broad regional divisions corresponding to west-central (Mali and Cameroon), eastern, and southern Africa (Fig. 2). Savanna elephants from eastern and southern Africa could be further differentiated geographically (Fig. 4). Within eastern African locations, the PCA inferred two clusters: one formed by Murchison Falls, Kidepo Valley, Eritrea and the other formed by Ethiopia, Amboseli, Central Kenya, Serengeti, Ngorongoro, Tarangire (Figs. 4 and S26). The locations in southern Africa formed distinctive southeastern (Mozambique, Kruger, Mashatu), south-central (Sengwa, Hwange, Zambezi, Chobe, Savuti) and southwestern (Etosha, Namib Desert) clusters (Fig. 4 and S26). The Admixture analysis converges until $K = 12$, at which point many locations form separate clusters (Figs. S27 and S28). It is important to note that the division between clusters (Fig. 4) may be inflated by gaps in sampling and the groupings that we describe are to a certain degree arbitrary. We hypothesize that future sampling from locations for which data are missing in our dataset will further contribute to the continuity of the isolation by distance pattern of genetic variation. However, we detect genetic similarity among geographically distant localities (e.g., Mali and Benoue/Waza in Cameroon) as well as clear genetic distinctions between geographically neighbouring locations (e.g., Serengeti and Tarangire), suggesting that our whole-genome data describes population structure with high resolution. Location-specific clusters are evident in PCA and Admixture even for populations with very low levels of genetic differentiation. For instance, in southeastern Africa, the Gorongosa National Park (Mozambique), Mashatu Game Reserve (Botswana), and Kruger National Park (South Africa) form separate genetic clusters in the PCA and Admixture, even though $F_{ST}$ is only 0.023 for each pairwise comparison of these populations (Fig. S5). Overall, $F_{ST}$ was low between pairs of savanna populations. Across pairwise comparisons, on average, $F_{ST}$ was 0.067 between pairs, excluding the elephants sampled at the two locations with a high proportion of hybrids in QENP (Ishasha and Mweya). Most genetic variation in savanna elephants exists within populations rather than between them, as also evidenced by the first two PCs in a PCA of savanna elephants (excluding QENP), based on called genotypes, explaining less than 29% of the variation in the data (Fig. 4).

To assess the regional levels of genetic diversity and inbreeding, we calculated the levels of genome-wide heterozygosity (Fig. 5, Supplementary Data 6, and Fig. S29) and ROH inbreeding coefficients (Fig. 6, Supplementary Data 7, and Fig. S30). In addition, to disentangle the impact of inbreeding and genetic drift on genetic diversity, we compared observed heterozygosity including and excluding the ROH regions (Fig. S13 and Supplementary Data 7).

Consistent with the weak population structure and with high gene flow, we found that heterozygosity and inbreeding are largely similar across the range of each species (Figs. 5 and 6 and Supplementary Data 6 and 7). The main deviation from the general pattern was in the western part of Africa. Savanna elephants from west-central Africa (Mali and Cameroon) and the forest elephant from Sierra Leone all had elevated levels of inbreeding (Fig. 6). This finding is further augmented when estimating the differences between heterozygosity, including and excluding ROH, which are highest in west-central savanna

elephants (Fig. S13), indicating a major impact of inbreeding in shaping elephant diversity in this region. This is consistent with the earlier and more severe human pressure in this region. West African elephants were affected comparatively earlier and more severely than populations in other regions of Africa[32,83], and thus, the populations had already collapsed before World War I[32,43]. As a result, the West African elephants survive in small, isolated populations for decades prompting questions as to their viability[32,84]. Contrary to expectations, the effects of isolation and small population size in west-central Africa were not clearly reflected in overall heterozygosity. While the west African forest elephant from Sierra Leone had low diversity ($H_O = 2.75 \times 10^{-3}$) relative to central African forest elephants ($H_O = 3.34 \times 10^{-3}$), the savanna elephants from Mali ($H_O = 1.74 \times 10^{-3}$) and Cameroon ($H_O = 1.27 \times 10^{-3}$ in Benoue and $H_O = 0.89 \times 10^{-3}$ in Waza) had higher genetic diversity than savanna elephants in other savanna regions. This pattern was due to the higher forest elephant ancestry in west-central African savanna elephants (Fig. S29).

A closer look at individual localities is revealing. The partitions in East Africa did not completely distinguish geographic localities, although Ethiopia and to some extent, Eritrea evidenced more discrete partitioning (Figs. 4 and S26–S28). Samples from Ethiopia are driving the variance on PC4 (Fig. S26) and form a separate cluster in the Admixture analysis from $K = 6$ (Figs. S27 and S28). For Eritrea, the differentiation is less clear because structure analyses generally struggle with singleton samples, which might have also influenced the surprising clustering with Uganda. The populations in the Babile Elephant Sanctuary of Ethiopia and in the Gash-Barka region of Eritrea are at the northeastern limit of the current savanna elephant range. Elephants in both locations were formerly connected to the broader species range, but went through bottlenecks in the early 1900s and now comprise small, isolated populations[85,86]. Moreover, the highest $F_{ST}$ in savanna elephants is between the Namib Desert and Ethiopia or Eritrea ($F_{ST} = 0.162$ and $0.156$, respectively), consistent with the great geographic distance between these locations, low within-locality diversity, and genetic drift likely driving divergence in small, isolated populations in different directions.

We also detected the effects of isolation in decreased levels of genome-wide heterozygosity (Figs. 5 and S20 and Supplementary Data 6). Savanna elephants in Eritrea ($H_O = 0.72 \times 10^{-3}$), Ethiopia ($H_O = 0.76 \times 10^{-3}$), and Namibia ($H_O = 0.80 \times 10^{-3}$ in Etosha and $H_O = 0.76 \times 10^{-3}$ in Namib Desert) have slightly reduced diversity compared to the other savanna populations. Furthermore, we found relatively high inbreeding in these isolated populations, perhaps due to reduced effectiveness of inbreeding avoidance mechanisms, as chances of finding a non-related mate are limited in small populations[87]. Interestingly, the reduced diversity was also evident outside of ROH (Eritrea: $H_O = 0.89 \times 10^{-3}$, Ethiopia: $H_O = 0.91 \times 10^{-3}$, and Namib Desert $H_O = 0.89 \times 10^{-3}$; Fig. S13), suggesting that low genetic diversity was driven not only by inbreeding, but also by genetic drift in bottlenecked populations. We analyzed the size distribution of ROH (Figs. 6 and S30 and Supplementary Data 7) and found that west-central African savanna elephants had the highest proportions of ROH in all size categories, suggesting constant long-term inbreeding; that elephants in Namibia have higher proportion of smaller ROH up to 5MB indicating mainly historical effects of inbreeding; and that elephants in Ethiopia and Eritrea have comparatively typical amount of ROH below 2MB but more of the large ROH categories, suggesting inbreeding in their more recent history. These patterns can be explained by their different population histories. In Ethiopia and Eritrea, elephants had been almost extirpated by the early 20th century[85,86], and in Eritrea, the population is still isolated by more than 400 km from the nearest population, which is reflected in its low genetic diversity[85]. In Namibia, the Namib desert has historically expanded and contracted, likely leading to a founder effect that limited the genetic diversity of the elephants[88].

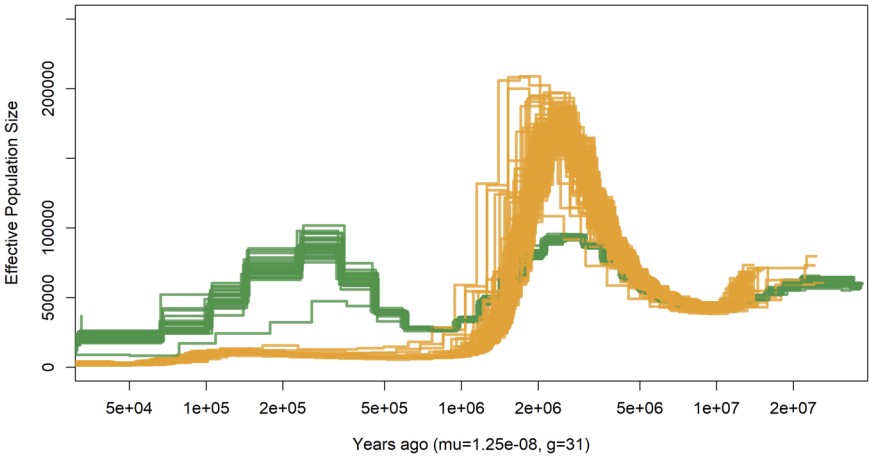

**Fig. 7 | Changes in effective population size over time estimated using PSMC (forest elephants in green and savanna elephants in orange).** Samples below 15× and hybrid populations (Garamba and QENP) were removed for clarity. Full versions of the plot are available in Fig. S31.

Thus, these results highlight the genomic consequences of isolation due to habitat loss and fragmentation, even in a species historically characterized by high mobility and gene flow. While the populations at the edge of the savanna elephant range (Ethiopia, Eritrea, Namibia) are to some extent extreme examples, habitat fragmentation and loss of genetic connectivity are threats even for historically large populations[89]. Our samples capture the status of African elephants in the 1990s and since then, habitat fragmentation and loss of connectivity has become an even more pressing issue[90,91]. In addition, the negative consequences of fragmentation may have been buffered by the elephants' ability to alter their behaviour in human-dominated landscapes in order to maintain connectivity[91]; however, elephants avoid landscapes with high human population density[37] and with the growing human population, maintaining connectivity and avoiding human-elephant conflict might become increasingly challenging.

By contrast, the savanna elephants across northern Botswana and Zimbabwe showed very little differentiation in the PCA and Admixture analyses (Fig. 4), and had some of the lowest $F_{ST}$ values in pairwise comparisons of localities within these countries ($F_{ST}$ ranged between 0 and 0.002, except for Sengwa with $F_{ST}$ ranging between 0.040 and 0.043). The low $F_{ST}$ indicates high genetic connectivity of elephants in the area and emphasizes the importance of maintaining connectivity through initiatives like the Kavango–Zambezi Transfrontier Conservation Area (KAZA; including Chobe, Hwange, Savuti, Sengwa and Zambezi in our dataset), one of the largest terrestrial nature and landscape conservation area in the world[92].

Strongholds of elephant density are in protected areas[41,93], and our data support the notion that where connectivity between protected areas is relatively preserved, e.g., in Kenya/Tanzania, in KAZA, and in Gabon/northern Congo, genetic diversity is maintained and inbreeding limited (Figs. 5 and 6). This highlights the importance of mitigating the isolation of protected areas[90] and ensuring that protected areas are effective in delivering positive conservation outcomes[94]. An evaluation of the effectiveness of protected areas in conserving African elephants showed that elephant population trends in protected areas were strongly associated with the extent of expenditure[95]. However, effective allocation of funds requires scientifically informed conservation strategies, which aid elephants and co-occurring species under their umbrella[96]. Using genomic data to identify genetic connectivity between populations, along with modelling habitat suitability[97], is a promising approach to identify areas that are important for maintaining elephant population connectivity.

## Genomic inventory of elephant diversity

We characterized the genetic diversity of each species (Fig. 5) and found that forest elephants consistently have much higher heterozygosity (mean $H_O = 3.46 \times 10^{-3}$) than savanna elephants (mean $H_O = 1.02 \times 10^{-3}$). Excluding the hybrid populations in QENP and Garamba leads to a slight decrease in observed heterozygosity (mean $H_O = 3.32 \times 10^{-3}$ in forest and mean $H_O = 0.87 \times 10^{-3}$ in savanna elephants). Thus, using whole-genome data and range-wide sampling, this confirms previous indications based on a limited number of nuclear markers[16,98] or a few individual genomes[17].

Several hypotheses have been offered for why the diversity of savanna elephants is lower than that of forest elephants, including ecological competition of savanna elephants with the elephantid *Palaeoloxodon* lineage, which became extinct towards the end of Pleistocene[99–101], or reduced effective population size due to high male-male competition in savanna elephants[22,102].

To better understand the patterns of genetic variation in savanna and forest elephants across time, we inferred the trajectory of effective population sizes using the pairwise sequentially Markovian coalescent (PSMC) method[103]. We observed that, as previously suggested[17], the savanna and forest elephants had very distinct trajectories after the split from the most recent common ancestor (Figs. 7 and S31).

It is important to take into consideration that the population trajectories might be influenced by unaccounted admixture events, as such events are not uncommon in proboscideans[17] while the timing of the population trajectories can be influenced by imprecise estimates of generation time and mutation rate. Even so, our results suggest that the split between the savanna and the forest elephant occurred around 4 million years ago (Mya), falling into the inferred split time of ≈2–5 Mya estimated by Palkopoulou et al.[17]. This approximately matches a period when open environments became more common in Africa[104,105], potentially providing a substrate for the ancestors of savanna elephants to expand to new habitats. Savanna elephant effective population size peaked around 3 Mya, which coincides with the African climate becoming periodically cooler and drier and arid-adapted biota spreading 2.8 Mya[106]. Interestingly, *P. recki*–a more specialized grazer[99]–is thought to have been present in large numbers in the open environments of East Africa at that time[100]. Thus, it remains unclear if African elephants reached high population sizes due to their dietary flexibility during the onset of open environments or dominated in other parts of the continent.

The effective population size of savanna elephants declined drastically after 2 Mya and remained low ever since (Fig. 7). We speculate that when the extent of open grasslands reached its peak

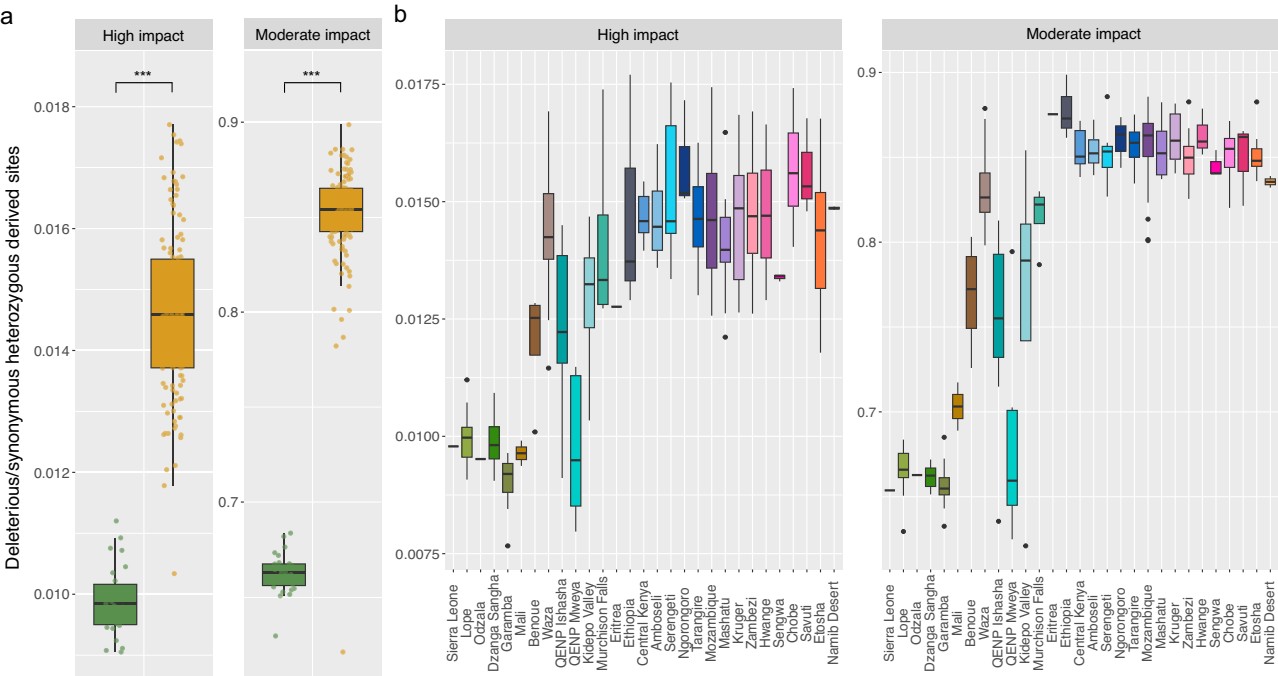

**Fig. 8 | Genetic load measured as the number of high impact (loss-of-function) and moderate impact (missense) heterozygous derived sites normalized by the heterozygous synonymous derived alleles as inferred in SnpEff v5.2f1[17].** Box-plots summarize the distribution of the data using the median (center line) and interquartile range (25th–75th percentiles; the box), with whiskers extending to 1.5× the interquartile range; outliers are shown as individual points. SnpEff results, along with individual and population information, are summarized in Supplementary Data 8. **a** Comparison between species (forest elephant in green and savanna elephant in orange). Populations with larger proportions of mixed ancestry (Garamba, QENP, Benoue, Waza, and Mali) were removed from the comparison. Normality was assessed using Shapiro–Wilk tests; two-sided Welch's t-tests were used for normally distributed data (high-impact variants), and Wilcoxon rank-sum tests were applied when normality assumptions were violated (moderate-impact variants). For high-impact variants, heterozygous load was significantly higher in savanna than in forest elephants (two-sided Welch's t-test: $t(101.1) = 28.83$, $p < 2.2 \times 10^{-16}$, $d = 3.70$, 95% CI [0.00441, 0.00506]). For moderate-impact variants, heterozygous load also differed significantly between species (Wilcoxon rank-sum test: $W(n_1 = 27, n_2 = 126) = 27$, $p = 1.16 \times 10^{-15}$, Hodges–Lehmann estimator = −0.191, 95% CI [−0.197, −0.185]). **b** Comparison across populations. The effect of admixture on load was visible in known hybrid locations (Garamba, Mali, Cameroon and Uganda). We also found that the isolated populations of Eritrea and Ethiopia show relatively elevated moderately deleterious variation and decreased highly deleterious variation, which is consistent with the high inbreeding levels and potential purging. In Mali, the pattern is coupled with the effects of mixed ancestry.

≈1.8 Mya[104], *P. recki* outcompeted the savanna elephant and pushed it to refugia until its extinction towards the end of Pleistocene[100]. It has been hypothesized that it was the more generalized diet of mixed-feeder/browser savanna elephant that allowed it to persist after the extinction of *P. recki*[99–101,107]. However, we caution that while our results roughly align with the palaeontological record, more data are needed to clarify whether other factors contributed to the low effective population sizes of savanna elephants.

On the other hand, the effective population size of forest elephants fluctuated, but remained higher than that of savanna elephants for the last one million years (Fig. 7), consistent with the detected higher genetic diversity (Fig. 5). We also investigated whether forest elephants harbour more genetic load than savanna elephants (Fig. 8). Recent studies in declining populations have indicated that populations that experienced a sudden decline from a previously large population size[108–110], are particularly exposed to negative fitness consequences due to fixation of harmful mutations, which had not been purged from their gene pool[111]. Due to the ongoing steep decline in forest elephant numbers[7,41], there is a concern that the decline might have fitness consequences. We used the data mapped to the Asian elephant to compare the rates of deleterious variation in savanna and forest elephants (Figs. 8 and S32 and Supplementary Data 8) using a functional annotation in SnpEff v5.2f[112]. We focused on the heterozygous load as a proxy of the masked portion of the genetic load[113,114], which reflects deleterious variation that does not affect the current generation, but has potential negative fitness effects in future generations if it becomes expressed. In addition, heterozygous load estimates are more robust to polarization and filtering bias. We found that the heterozygous genetic load of high- and moderate-impact sites (Fig. 8) is significantly larger in savanna compared to forest elephants. Statistical significance was assessed using two-sided Welch's t-tests (high-impact variants; $t(101.1) = 28.83$, $p < 2.2 \times 10^{-16}$, $d = 3.70$, 95% CI [0.00441, 0.00506]) and Wilcoxon rank-sum tests (moderate-impact variants; $W(n_1 = 27, n_2 = 126) = 27$, $p = 1.16 \times 10^{-15}$, Hodges–Lehmann estimator = −0.191, 95% CI [−0.197, −0.185]). The observation that forest elephants do not have an excess of deleterious mutations can be considered a positive outcome for the short-term survival of the species. The effect of admixture on load was visible in known hybrid locations (Garamba, Uganda, Cameroon and Mali). We also observed that the isolated populations of Eritrea and Ethiopia show relatively elevated moderately deleterious variation and decreased highly deleterious variation, which is consistent with the high inbreeding levels and inbreeding depression leading to purging. In Mali, this pattern is coupled with the effects of mixed ancestry.

In Summary, historical gene flow within and between species has been an important evolutionary process in African elephants. Hybridization between species has been a part of the evolutionary history of elephants and their extinct relatives, such as mammoths and straight-tusked elephants[17,71,72]. We found evidence that, although restricted,

gene flow between savanna and forest elephants occurred at several points in time since their evolutionary lineages diverged, with most savanna populations carrying trace amounts of forest ancestry in their genomes.

Within species, we found evidence of long-distance gene flow maintaining low levels of differentiation with genetic connectivity stabilizing levels of genetic diversity across large distances in eastern and southern Africa. In particular, elephants in Kavango–Zambezi Transfrontier Conservation Area, the largest population of elephants in the world, appear to be practically panmictic and they do not show the reduced diversity or high inbreeding of smaller, more isolated populations, highlighting the importance of protecting large and well-connected elephant habitats.

While we show that until recently, gene flow was an important process in elephant evolution, we are also seeing genomic consequences of centuries of anthropogenic pressure, which is only going to become more pronounced with the growing human population and habitat conversion, as well as ongoing poaching for ivory. As the increasing fragmentation of habitats hinders gene flow, the loss of genetic connectivity is becoming a major concern[89]. We found evidence of genetic drift in bottlenecked, isolated populations at the extremes of the savanna elephant distribution, such as in Ethiopia and Eritrea, even though our samples originate from the 1990s, and in a long-lived mammal, do not capture the changes driven by the most recent anthropogenic activities of the last 30 years. This has important genetic consequences, as small populations are more exposed to stochastic extinction factors and the extinction vortex[115,116].

In addition, accumulation of deleterious variation in small populations can lead to mutational meltdown[117], because populations with high effective population size ($N_e$) that suffer sudden bottlenecks are particularly affected by the fixation of harmful mutations that had not previously been purged from their gene pool. This would be a concern for the forest elephant, due to its high historical effective population size coupled with drastic population decline in the last decades. However, our estimation that forest elephants have significantly lower heterozygous load than savanna elephants is a very positive finding for the short-term survival of the species.

Our findings provide a map of the genomic diversity of African savanna and forest elephants, which shall facilitate further research and conservation actions. First, this dataset can serve as a publicly available reference against which any future studies of individuals or populations can be compared, including when assessing population structure or levels of genetic diversity. Second, by describing the genetic composition of each species, this dataset lays the groundwork for defining conservation units as a prerequisite of future conservation actions. Third, as our samples originate from the 1990s and pre-date the most recent population declines by about one elephant generation, this data also serves as a baseline for tracking more recent changes. Estimates of inbreeding from runs of homozygosity in the genome are a particularly powerful way to track even recent effects of population declines. Fourth, our results highlight the formative effect of gene flow on patterns of genetic diversity in African elephants. We highlight populations that show evidence of genetic isolation and have relatively low genetic diversity, as well as those where genetic connectivity seems relatively preserved and can be maintained by continued conservation efforts. This illustrates that genomic estimates of gene flow, especially in conjunction with habitat suitability modelling, can be used to inform conservation management. Lastly, the broad geographic span and high quality of the sampled genomes offer the opportunity to identify SNPs that are informative regarding individual identification, relatedness, species identification, and geographic provenance, which can be used in forensics or population monitoring. Overall, by informing conservation of elephants, ecosystem engineers and keystone species, our findings can have a far-reaching impact on the African ecosystem.

## Methods

### Ethics and inclusion statement

The data generated in this study originates from biobanked samples collected for previously published research that was performed primarily in the 1990s. Researchers and sample providers who were involved in the sample collection or in the subsequent steps are credited in Acknowledgements or have co-authored this study if they meet the requirements for authorship. Authorship roles were communicated transparently and approved. Utilizing biobanked samples allowed us to avoid affecting wild populations, eliminating concerns for animal welfare and environmental protection. The biobanked samples are deposited in collections at the University of Illinois at Urbana-Champaign in the USA (UIUC) and at the University of Copenhagen in Denmark (UCPH). Samples were obtained in full compliance with required CITES and other national and international permits. Samples were imported from Uganda to Copenhagen, Denmark, under CITES import permit IM 1208-989/05. The experiments in the USA were conducted under the University of Illinois Institutional Animal Care and Use Committee approved protocol number 24029 and CITES permits US 750138 and US 756611. The samples were collected before the Nagoya Protocol came into effect. While the samples predate current benefit-sharing frameworks, we recognise the importance of equitable use of biodiversity resources. All sequencing data generated in this study is now freely accessible in a public repository, the European Nucleotide Archive. Generating a range-wide dataset mapping the genomic diversity of elephants across the African continent streamlines future study of local elephant populations on a finer scale and facilitates the use of genomics in in-situ research and conservation. This study is co-authored by researchers and conservationists at institutions in range countries, who have been instrumental in sampling, putting the results into context, and in delivering interpretations that advance our knowledge about elephants, complementing the broad body of published work cited in this study. By making the research open access, we aim to promote equitable access to insights derived from these samples. Through communication with local stakeholders, we strive to make the data relevant for elephant conservation and local capacity building.

### Samples, DNA extractions, and sequencing

We analyzed a total of 249 genomes (Supplementary Data 1) representing individuals from the two species of elephants currently living in Africa, the African savanna elephant (*Loxodonta africana*) and the African forest elephant (*L. cyclotis*). Of the 249 genomes, 226 were newly sequenced here, and 24 were published previously[17,51,52]—this includes one sample (DS1546), both published previously[17] and resequenced here. The two sample sets at UIUC and UCPH were processed and sequenced separately, but analyzed jointly (Supplementary Data 9). In addition, two previously published genomes[17] of the Asian elephant (*Elephas maximus*) and one previously published[118] genome of the woolly mammoth (*Mammuthus primigenius*) were included in the dataset for analyses for which outgroups were needed.

Wild elephants were primarily sampled by biopsy darting as previously described in ref. 119, some samples were obtained from blood, dry skin, scrapings from carcass remains, or dung.

Library preparation and sequencing was performed in several rounds (Supplementary Data 6). DNA extractions and lower coverage sequencing (≈1–15× coverage) was performed at UIUC and UCPH. Additional sequencing at high depth was performed by Illumina Laboratory Services targeting 40× coverage.

### Samples at the University of Illinois

DNA samples had been previously extracted or were newly re-extracted from stored samples at UIUC. The DNA was isolated using the Wizard Genomic DNA Purification Kit (Promega, Madison, USA), DNeasy Blood & Tissue Kits (Qiagen, Hilden, Germany), or QIAamp

PowerFecal Pro DNA Kit (Qiagen). DNA samples were quantified using a Qubit fluorometer.

**Roca_batch0.** For 20 forest elephant samples, 100 ng of DNA was submitted to the Roy J. Carver Biotechnology Center at UIUC for initial screening. Libraries were constructed using the TruSeq library kit (Illumina, San Diego, USA) and sequenced on two lanes of 2 × 75 bp reads on the HiSeq 4000 platform.

**Roca_batch1.** Subsequently, 165 samples, including the 20 forest elephant samples from Roca_batch0, were submitted to the Roy J. Carver Biotechnology Center at UIUC to sequence 5 lanes of 2 × 150 bp reads on NovaSeq6000 platform S4 flow cells. The same libraries were used for the 20 forest elephant samples, while for the new 145 samples, 100 ng of DNA was submitted and libraries were prepared by using Kapa shotgun kits (Roche, Basel, Switzerland). We targeted either low coverage of 4× or higher coverage of 15×.

**Illumina_batch1-3.** Finally, 167 samples were sent to Illumina Laboratory Services (San Diego, USA) for a targeted 40× coverage sequencing on the NovaSeq6000, including some of the samples sequenced previously to lower depth in Roca_batch0 and Roca_batch1. Paired-end (PE) libraries for all samples were generated using the TruSeq DNA PCR-Free Sample Prep kit (Illumina) or the DNA PCR-Free Prep kit (Illumina), depending on the amount of sample available. Samples with >110 ng of DNA available were processed with the standard Illumina DNA PCR-Free kit protocol, and those with 25–109 ng of DNA with the low-input protocol. Any samples with <25 ng of DNA were prepared with the Illumina DNA Prep kit. All protocols followed the manufacturer's instructions.

Pre-fragmentation gDNA cleanup used paramagnetic sample purification beads. Sample DNA was fragmented, and libraries end-repaired prior to size selection using TruSeq DNA PCR-Free beads for even coverage of areas known to be challenging to sequence. Libraries were evaluated for yield using a real-time qPCR assay, and Illumina DNA Standards with primer master mix qPCR kit (KAPA Biosystems, Roche). Following library quantitation, DNA libraries were denatured, diluted and clustered onto patterned flow cells, and then normalized to 1.5 nM for the NovaSeq6000 prior to clustering and sequencing. DNA libraries were denatured, diluted and then loaded onto patterned flow cells. Subsequently, flow cells were placed on a NovaSeq6000 instrument, where onboard clustering and sequencing occurs on 151 bp paired-end, indexed runs for pooled libraries for a targeted 40× coverage. Sequencing runs followed the NovaSeq6000 Sequencing System Guide. Illumina NovaSeq Control Software v1.7.5 (NVCS) was used with the NovaSeq6000 sequencers as well as Real-Time Analysis. FASTQ files were automatically generated in the NovaSeq workflow and downloaded from BaseSpace.

### Samples at the University of Copenhagen
Thirty tissue samples were collected in Queen Elizabeth National Park (QE: sector Ishasha−IS and sector Mweya -MW), Kidepo Valley National Park (KV) and Murchison Falls National Park (MF). To avoid sampling closely related individuals, individuals were randomly sampled from different family groups, while individuals from MF were solitary males.

DNA was extracted following the manufacturer's protocol instructions of the QIAGEN DNeasy Blood & Tissue Kit (QIAGEN, Valencia, CA, USA). RNase was added to all samples to ensure RNA-free genomic DNA. DNA was eluted in a solution of 10 mM Tris-HCl, 1 mM EDTA (1 M TE, pH 8.0), then stored at 4 °C. DNA concentrations were then measured using a Qubit 2.0 Fluorometer and Nanodrop before using gel electrophoresis to check the quality of genomic DNA.

**Uganda_batch1-2.** Sequencing libraries were prepared from the DNA extracts and sequenced at the GeoGenetics Sequencing Core

in Copenhagen, Denmark in paired-end 2 × 150 bp mode on three lanes of the Illumina NovaSeq6000 S4 platform, targeting ≈15× coverage.

In the analyses, we treat Ishasha and Mweya as separate locations despite their close proximity as some previous work suggests that the Kazinga Channel might be a barrier between the populations[120,121]. In addition, these populations were strongly affected by the civil conflict and associated population decline in the 1970s–1980s[122].

### Mapping
Sequencing reads were processed and mapped using a development version of the Paleomix pipeline (Schubert et al., 2014) available at https://github.com/MikkelSchubert/paleomix (branch "pub/2022/africa") against (a) the African savanna elephant LoxAfr4 genome assembly[17] combined with the mitochondrial reference genome sequence available from GenBank (NC_000934.1); and (b) the Asian elephant genome assembly mEleMax1 generated by the Vertebrate Genomes Project and available from NCBI under accession no. PRJNA850184.

This mapping pipeline[123,124] included trimming of adapters and merging of overlapping reads using AdapterRemoval v2.3.1[125]. Mapping was performed using the BWA v0.7.17[126] "mem" algorithm. No quality trimming or filtering was performed during these steps, excepting that reads containing no genomic DNA (primer dimers) were filtered prior to mapping. Alignments were normalized, sorted, and merged using the SAMtools v1.13[127] "fixmate" with option "-m", "calmd", "sort", and "merge" commands. Putative PCR duplicates were flagged using SAMTools "markdup" for unmerged reads and the "paleomix rmdup_collapsed" tool for merged reads. The final BAMs were filtered to remove paired alignments spanning contigs or in improper orientations, alignments with insert sizes not in the range 50–1000 bp, or with fewer than 50% or fewer than 50 bp of the query sequence mapping to the reference genome. Additionally, unmapped reads and partially mapped pairs, secondary alignments, QC-flagged reads, PCR duplicates, and supplementary alignments were filtered.

### Reference filtering
We performed a rigorous filtering of genomic sites to avoid analyzing error-prone regions, following a previously developed pipeline[123,128]. Both reference genomes used in this study are chromosome level and only autosomal sites were used in the analyses. The filtering steps resulted in a final dataset of 1,216,996,995 sites in the African savanna elephant reference (LoxAfr4) and 1,330,441,632 sites in the Asian elephant reference (mEleMax1).

**Repeat masking.** We identified repetitive and low-quality regions in both reference genomes using RepeatModeler v2.0.1[129] and RepeatMasker v4.1.0[130]. The database rmblast v2.10.0[131] was used to build the database. RepeatMasker was run with the -gccalc option.

**Mappability.** We evaluated the mappability of each site in the reference genomes using GEM v20130406-045632[132]. The mappability was estimated with kmers of 150 bp and allowing for two mismatches and two differences (-m 0.0134 -e 0.0134 -l 150). Only sites with a mappability of 1 (i.e., uniquely mapping) were retained.

**Global depth.** We calculated the global sequencing depth across all samples using SAMtools v1.13[126], and then removed sites with extremely low (half of the median) and high (1.5-times the median) global depth.

**Excessive heterozygosity.** We filtered out potentially paralogous regions by identifying sites with excessive heterozygosity as inferred by estimating per-site inbreeding coefficients (F) in PCAngsd v0.99[133,134]. Windows of 10 kb surrounding the sites exhibiting

excessive heterozygosity ($F < −0.90$) and significant deviations from Hardy-Weinberg equilibrium ($p < 10^{-6}$) were excluded.

## Sample filtering

We used the QC pipeline, building on the mapping pipeline[124,135] to generate statistics about the mapping. From the total dataset of 249 African elephant samples, we excluded 17 samples that were flagged in our quality checks due to low mapping rate, low coverage, relatively high error rate, high GC content, extreme levels of heterozygosity, and relatedness (Supplementary Data 2).

**Identification of closely related or duplicated samples.** We identified duplicated or closely related individuals using R0, R1, and KING-robust kinship[136,137] based on 2D SFS spectra estimated in ngsRelate v2[138].

Ten samples were identified as duplicates or first-degree relatives. For individual-based analyses, the eight first-degrees relatives were included, but they were excluded in population-based analyses (Supplementary Data 3).

## Datasets

**Genotype calls.** Genotype calls for the 207 high-coverage genomes were generated jointly following the genotype calling pipeline described in ref. 124 and using BCFtools v1.17[127] with the "−per-sample-mF" flag in "bcftools mpileup", the "−multiallelic-caller" flag in "bcftools call", and a minimum base quality of 25 and mapping quality of 30. The called sites were filtered to only represent the reliable sites inferred in the Reference filtering section, and to remove indels and multiallelic sites. For some analyses (Supplementary Data 3), we furthermore masked sites with read depth below 10 and heterozygous calls with less than two reads supporting the minor allele.

**Imputation and phasing.** We performed imputation and statistical phasing on 232 samples (both savanna and forest elephant individuals) using BEAGLE v3.3.2[139] from genotype likelihoods (GLs) estimated in ANGSD v.0.934[140]. GLs were estimated using the GATK genotype likelihood model (-GL 2) and only keeping sites that had a $p$-value less than $1e^{-6}$ (-SNP_pval 1e-6) for being variable in addition to only keeping sites that passed initial QC (-sites) as well as using a minimum MAF of 0.05 (-minMAF 0.05). We assumed the major allele was fixed and the minor was unknown when estimating GLs (-doMajorMinor 1 -doMAF 2). We further filtered the imputed results by only keeping sites with an imputation score $R^2 > 0.95$ and no missingness permitted after applying a > 0.95 posterior probability cutoff on genotype calls. We used the phased version of our imputed results.

## Population structure

**PCA.** A PCA was performed on the dataset of imputed genotypes in all 232 genomes, and using called genotypes for the separate savanna and forest datasets using PLINK v.1.9[54].

Since PC1 clearly captured the gradient of savanna vs forest elephant ancestry, we used PC1 as one of the measures to evaluate forest ancestry in savanna elephants (Supplementary Data 4). We used a savanna elephant with the lowest PC1 value (KRSWAZ) and a forest elephant with the highest PC1 value (LO3517) and calculated the forest ancestry for each elephant sample as: 1-([sample]-KRSWAZ)/(LO3517-KRSWAZ).

**Admixture and evalAdmix.** We used the imputed genotypes to estimate admixture proportions in three different subsets of the dataset. One subset included the two species of elephants, and two additional species-specific subsets to better resolve the within species population structure. After excluding a sample of each first-degree relative and keeping only SNPs with minimum MAF of 0.05, we kept 224, 176, and 48 individuals and 7,556,357, 2,348,778 and 11,549,205 SNPs for the

subset with both species, the subset of savanna elephants and the subset of forest elephants, respectively. We used ADMIXTURE v1.3.0[55] to estimate admixture proportions, and starting at $K = 2$ we sequentially tried larger $K$ values until reaching a value that could not reach convergence across different independent runs. We considered it converged when out of a minimum of 5 independent runs and a maximum of 100, the lowest difference between an admixture proportion estimate was below 0.001 among the three runs with the highest final maximum likelihood, using for the comparison the permutation of columns that minimized the root mean square error between the Q matrices of admixture proportions.

We obtain convergence for up to $K = 6$, $K = 12$, and $K = 3$ in the subsets with both species, with savanna elephants and with forest elephants, respectively. For each converged $K$ in each converged subset of data, we used evalAdmix v0.962[56] to evaluate the model fit of the inferred admixture proportions of the maximum likelihood run.

**IBS tree.** To construct a neighbor joining (NJ) tree (Fig. S33), we estimated an identity-by-state (IBS) matrix as a measure of the pairwise genetic distance between individuals, using the command "−distance" in PLINK v1.9.0[54]. We further inferred the NJ tree using the R[141] package ape[142].

**Maximum-likelihood tree.** We constructed an autosomal phylogeny that included all 207 high-coverage samples in our study and an Asian elephant for the outgroup (Fig. S17). We used the VCF file described above (Datasets: Genotype calls), excluding sites with read depths below 10 and heterozygous calls with less than two reads supporting the minor allele. We extracted 100,000 randomly-selected SNPs using the command "shuf". We converted the VCF file to a phylip file format using vcf2phylip.py (https://github.com/edgardomortiz/vcf2phylip). Next, we inferred a maximum-likelihood phylogenetic tree using the 100,000 autosomal SNPs using IQ-TREE v1.6.12[68], where we estimated the best model using ModelFinder[69] and used 1000 ultra-fast bootstraps to infer each tree. We also inferred savanna-specific (Fig. S18) and forest-specific (Fig. S19) elephant autosomal phylogenetic trees, in which we subsetted the VCF to include only savanna and forest elephants with an Asian elephant outgroup, respectively, and followed the same methods as inferring the autosomal phylogeny with all samples included, with the exception of using 50,000 autosomal SNPs for the forest elephant dataset.

**EEMS.** We ran an Estimation of Effective Migration Surfaces (EEMS; Petkova et al.[143]) to assess connectivity between populations of the same species. A matrix of identity-by-state (IBS) distances between pairwise samples was generated using the "bed2diffs" script, which is integrated into the EEMS workflow. We set the analysis to use 500 demes for geographic resolution. We ran three independent Markov chain Monte Carlo chains, with 30 million iterations, including a 15 million iteration burn-in period. The final EEMS output was visualized with a custom R script from the reemsplot package (http://www.github.com/dipetkov/eems).

**qpGraph.** To explore admixture graphs, we performed analyses using the "find_graphs" function in the ADMIXTOOLS2[60] package, and the approach described in refs. 124,136. Briefly, the "extract_f2" function was first used to extract f2-statistics for each population using block lengths of 4 Mb. SNPs with a maximal missingness of 0.9 within any given population were retained. We subsequently ran "find_graphs" to explore admixture graphs with the assumed number of admixture events, from 0 to 5. A testing procedure was subsequently applied, identifying the best-scoring graph out of the 500 candidate graphs for a given number of admixture events, and identifying the graphs with the same number of admixture events with fit scores statistically indistinguishable to the best-scoring graph. In this process, a test score

was calculated by optimising a topology from a subset of the *f*-statistics and comparing this with the remaining graphs. Significance was assessed using a jackknife approach for each obtained graph, wherein each graph was compared to the remaining graphs using a nominal *P*-value of 0.05. Utilising a similar process, we also tested whether the best-scoring graph for each number of admixture events was significantly different to a graph with a higher number of admixture events. Subsequently, the optimal number of admixture events was determined by choosing the set of graphs with a lower number of admixture events when compared to a set of graphs with a higher number of migrations, and which were not significantly different. The results are available in Fig. S15.

**Treemix.** We used TreeMix v1.13[67] to estimate population splits and migration events using the imputed data for 224 elephant samples. Our dataset consisted of 7,556,357 SNPs after excluding a sample of each first-degree relative and keeping only SNPs with minimum MAF of 0.05. We clustered elephant samples into 7 populations: 4 populations within the African savanna elephant (SavannaEast, SavannaSouthCentral, SavannaSouthEast, SavannaSouthWest) and three populations within African forest elephants (ForestCentral, ForestEast, ForestWest). We first inferred a maximum likelihood tree of the populations and generated bootstrap replicates by resampling blocks of 500 SNPs. We inferred an initial tree with no migration events and subsequently estimated 1–7 migration edges (Fig. S16).

**F$_{ST}$.** As part of the genotype calling pipeline[123], we generated site frequency spectra (SFS), and two-dimensional SFSs to estimate $F_{ST}$ using the corrected Hudson's estimator[144]. First-degree relatives were excluded.

**D-statistics.** We used the called genotypes on the high-depth samples to estimate D-statistics[145]. We used the qdpstat function in ADMIXTOOLS2 v.2.0.0[60], with a block size of 5 Mb to estimate standard errors with the built-in block jackknife approach. Driven by observation of a low but detectable proportion of forest ancestry in some savanna populations in Admixture analyses, we wanted to further investigate potential admixture of forest elephants into savanna elephant populations. We conducted grouping of samples of the form (Savanna1, Savanna2, Forest, AsianElephant), where Savanna1 and Savanna2 correspond to two samples of savanna elephant from different locations, and Forest to a sample of forest elephant. We tested samples from all locations of savanna elephants as H1 and H2, and all locations of forest elephants except Garamba due to the presence of recent savanna elephant admixture. Next, we performed the same analyses in the other direction, placing all forest elephant populations in H1 and H2, while as H3, we put either Kruger (as the geographically most distant population from the tropical forest) or all relatively non-admixed savanna populations.

**F4 ratios.** To estimate the ancestry proportions in each admixed individual, we calculated F4 admixture ratios using "qpadm" implemented in ADMIXTOOLS2[145]. This models a target population as a mixture of two source populations, given a set of outgroup populations[146]. The F4 ratio test estimates admixture proportion ($\alpha$) in a five-population scenario as $\alpha$ = f4(A, O; X, B) ÷ f4(A, O; C, B), including a test population (X; all savanna populations), an outgroup (O; Asian elephant), a reference "unadmixed" savanna population (B; Kruger), a reference "unadmixed" forest population 1 (A; Sierra Leone as west forest population), and a reference "unadmixed" forest population 2 (C; Lope combined with Dzanga Sangha as central forest population). In other words, we estimated F4 ratios in the form of $\alpha$ = f4 (Sierra Leone, Asian; target, Kruger)/f4 (Sierra Leone, Asian; Lope/Dzanga Sangha, Kruger). We used $5 \times 10^6$ as the SNP block size for jackknifing.

We tested the correlation between F4 ancestry and geographic distance-to-tropical forest. We used the WWF Terrestrial Ecoregions of the World shapefile[147] to extract and merge polygons corresponding to Congo-Guinean rainforests, and then we calculated great-circle distances to the nearest rainforest boundary. Pairwise Euclidean distance matrices were then computed for (i) ancestry proportions and (ii) distance-to-rainforest values, with an additional matrix of straight-line geographic distances between sampling sites. Finally, we applied the Mantel test (with Spearman correlation and 9999 permutations) in R[141] v4.4.3 using the package vegan v2.7-2[148] in RStudio v2024.12.1.563[149].

**apoh.** We explored the presence of recent admixture between forest and savanna population using *apoh* (commit 3f6491fa55d9820 f558368ecfc97eac4e1277a0f)[59]. We used the imputed dataset and the ancestral population allele frequencies estimated with ADMIXTURE with all elephants assuming $K = 5$ to estimate parental ancestry proportions using NGSremix v1.1.0[150], selecting for the analyses individuals that had at least a 0.025 proportion of ancestry deriving from populations corresponding to both elephant species. We then used *apoh* to test for the presence of recent admixture and reconstruct recent admixture pedigrees that best explained the observed patterns of ancestry. We define as "recently admixed" samples those for which a pedigree where parents have different ancestry proportions fits better the observed paired ancestries than a pedigree where all parents have the same ancestry proportions.

## Demography and genetic diversity

**Runs of homozygosity (ROH).** ROH analyses were performed using PLINK v1.9[54]. PLINK files were generated from the imputed dataset, including all 232 individuals (including 1st and 2nd degree relatives). We excluded non-variable sites, SNPs with <95% genotyping rate (−missing −geno 0.05), and SNPs with MAF < 0.05 (−maf 0.05). Filtering based on depth and heterozygous calling was not applicable due to the imputation. Twenty-seven complete autosomal chromosomes were analyzed spanning 2.6 Gb. For each individual, we called ROH by using the default settings (−homozyg) while allowing for 3 heterozygous sites (−homozyg-window-het 3), and 20 missing sites (−homozyg-window-missing 20) per scanning window.

**Heterozygosity.** We calculated genome-wide heterozygosity as the proportion of heterozygous sites for each sample based on genotype calls. We included the genotype calls at both variable and non-variable sites, where we masked all genotypes supported by fewer than ten reads or exceeding twice the global average depth and heterozygous genotypes supported by fewer than two reads for any of the two alleles. To account for the impact of recent inbreeding on levels of genetic diversity, we also calculated genome-wide heterozygosity outside ROH. For each sample, we masked its identified ROH regions and only estimated the proportion of heterozygous sites in the remaining genomic regions, following the same filtering criteria.

**PSMC.** We used the PSMC[103] to estimate population size trajectories across time for 207 high-depth individuals. For each sample, we called genotypes individually using BCFtools v1.17[151], using the full BAQ model in the mpileup step and the consensus caller in the call step. For each sample, we set to missing sites where the depth was lower than ⅓ of the sample average depth (or to 6 for samples where ⅓ of the average depth was lower), sites were the depth was higher than 2 times the average depth of the sample, and sites called as heterozygous with less than 2 reads supporting either of the two alleles. We then generated the PSMC input files using the default settings, except setting the time intervals to "1 + 1 + 1 + 1 + 25*2 + 4 + 6" to get a better resolution in younger times and avoid artifactual recent size changes[152].

**popSizeABC**. To estimate historical population sizes, we employed the popsizeABC[153], which utilizes Approximate Bayesian Computation (ABC) to infer demographic history in two populations of forest elephants that had genomic data for a sufficient number of individuals (Lope and Dzanga Sangha). This method integrates coalescent simulations and statistical models to approximate the posterior distribution of effective population sizes over time. For each of selected populations, we provided popsizeABC with input parameters derived from whole-genome sequence data, including allele frequency spectra and summary statistics such as nucleotide diversity and linkage disequilibrium patterns. The ABC algorithm was implemented with 210,000 iterations to ensure convergence and accurate posterior estimates. We used a tolerance threshold of 0.1 to accept simulations closely matching the observed data. We set the generation time as 31 years following previous studies[17,154], which used the estimate of 31 years as an average of 17-20 years in females[155,156] and 40–49 years in males[102,157]. The generation time of 31 years also matches the current female generation time estimate for forest elephants[158], which is slightly longer than the current female generation time estimate of ~25 years in savanna elephants[159,160]. Finally, we visualized the inferred population size changes over time to identify key historical demographic events (Fig. S34).

### Genetic load

We estimated genetic load following the pipeline in Dussex et al[160]. available at https://github.com/ndussex/Crater_lion_genomics/tree/main. Briefly, we created an ancestral-state fasta file by merging two bam files representing outgroup species—an Asian elephant from Myanmar[17] and a Pleistocene woolly mammoth, NSI12.8K[118]—downsampled to the same genomic coverage, on which we called consensus using ANGSD[140] -doFasta 2. We then replaced any site in the Asian elephant reference genome (mEleMax1), where the ancestral sequence differed using the provided script[161].

We used SnpEff v5.2f[112] to obtain functional annotation. First, we used gffread v0.12.8[162] to filter the gff annotation file provided with the mEleMax1 genome assembly for missing START or STOP codons, in-frame stop codons (-J option), and those with pseudogene attributes (−no-pseudo). We split a VCF with data mapped to the Asian elephant into one VCF per sample using BCFtools v1.17[151] and we annotated variants in each VCF file separately. We used SnpSift v5.2f[163] to filter variants per effect and zygosity. We used a custom script to collect the counts, following the "high", "moderate", and "low" impact categories inferred by SnpEff[112]. In addition, we collected the synonymous variants ("synonymous_variant") as a separate category and used this for normalization. We plotted the results in RStudio v2024.12.1.563[149].

We compared the genetic load in savanna and forest elephants and we followed the definition by Bertorelle et al[113], which splits the total genetic load into realized load (reducing fitness in the current generation) and masked load (fitness effects are hidden but may become expressed and lead to potential reduction of fitness in future generations). In the absence of direct estimates of selection coefficients and dominance effects, we used homozygous load as a proxy for realized load and heterozygous load as a proxy for masked load[114]. We define homozygous load as the number of homozygous deleterious (high-impact or moderate-impact) derived sites normalized by the number of homozygous synonymous derived sites, and we define heterozygous load as the number of heterozygous deleterious (high-impact or moderate-impact) derived sites normalized by the number of heterozygous synonymous derived sites. When interpreting the results, we focused on the heterozygous load (Fig. 8), which is more robust to biases in filtering and polarization, and is also the most relevant when assessing the potential future fitness effects. We show additional measures of load in Fig. S32. When comparing the heterozygous load in savanna and forest elephants, we performed statistical comparisons in R v4.5.1 using package effsize[164]. Normality was assessed using Shapiro−Wilk tests; two-sided Welch's *t*-tests were used for normally distributed data (high-impact variants), and Wilcoxon rank-sum tests were applied when normality assumptions were violated (moderate-impact variants).

### Reporting summary

Further information on research design is available in the Nature Portfolio Reporting Summary linked to this article.

## Data availability

The sequencing data generated in this study have been deposited in the European Nucleotide Archive database under accession code PRJEB77639. See Supplementary Data 10 for a complete list of accession codes. Previously published data were accessed from the European Nucleotide Archive under accession codes: PRJNA622303[51], PRJEB24361[17], PRJNA761769[52], PRJEB59491[118]. The data was mapped to the following reference genomes: Asian elephant mEleMax1 available from NCBI project PRJNA850184, mitogenome available from GenBank ID NC_000934.1 [https://www.ncbi.nlm.nih.gov/nuccore/NC_000934.1], and African savanna elephant Loxafr4 available from ftp://ftp.broadinstitute.org/pub/assemblies/mammals/elephant/loxAfr4/.

## Code availability

Code used to analyse the data is available at https://github.com/patriciapecnerova/STAMPEDE.

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

## Acknowledgements

P.P. received funding from the European Union's Horizon 2020 research and innovation programme under the Marie Skłodowska-Curie grant agreement No. 892446, the L'Oréal-UNESCO For Women in Science award, National Geographic Society grant (EC-93655C-22), the Branco Weiss Fellowship—Society in Science, and FORMAS grant no. 2023-01441. M.S. was supported by the Novo Nordisk Foundation grant NNF23SA0084103 for the Center for Basic Metabolic Research, Copenhagen University. L.M.H. thanks the National Science Foundation Postdoctoral Research Fellowship (award number 2208950) for funding and support. Illumina Inc. provided support for next-generation library sequencing. For further funding, we thank the US Fish and Wildlife Service African Elephant Conservation Fund, grant F22AP01215; A.L.R. also thanks Fulbright Denmark and the UIUC College of ACES Seed Grant program. The authors acknowledge the use of computing resources at the core facility for biocomputing at the Department of Biology, University of Copenhagen. The authors would like to acknowledge the contribution of Yirmed Demeke, who provided the Ethiopian samples, and would like to thank Peter Arctander for help with collecting samples from Uganda. P.P. would like to thank Love Dalén for useful discussions on initial conceptualization of the study design. The authors acknowledge the contribution by David Reich to screening genomes of forest elephants. The authors thank David Díez del Molino for providing access to woolly mammoth VCF and BAM files for the genetic load analysis.

## Author contributions

P.P., R.H., H.R.S. and A.L.R. designed the study. P.P., H.R.S. and A.L.R. provided funding. G.W., I.D.-H., P.J.V.C.d.G., V.B.M., C.M., N.J.G., H.R.S. and A.L.R. collected or provided access to samples. Y.I., A.A.-C., S.U., K.N., S.T. and K.V.-M. administered laboratory analyses. P.P., G.G.-E., L.D.B., C.G.S., X.L., A.B.-O., A.K., L.M.H., R.F.B., L.L., M.S.R., X.W. and M.S. performed genomic analyses and contributed scripts. Y.I., A.d.F., R.S.M. and A.L.R. provided sequencing data. A.d.F., Sa.B., N.T., M.W., M.N.T., St.B., C.T., I.D.-H. and G.W. consulted the results and provided advice on biological interpretations. P.P. wrote the manuscript with input from C.H., R.H., H.R.S. and A.L.R. All authors contributed to and approved the final manuscript.

## Funding

## Competing interests

The authors declare no competing interests.

## Additional information

[1]Section for Computational and RNA Biology, Department of Biology, University of Copenhagen, Copenhagen, Denmark. [2]Department of Biology, Lund University, Lund, Sweden. [3]Copenhagen Zoo, Frederiksberg, Denmark. [4]Department of Animal Sciences, University of Illinois at Urbana-Champaign, Urbana, IL, USA. [5]Bioinformatics Research Center, Department of Molecular Biology and Genetics, Aarhus University, Aarhus, Denmark. [6]National Centre for Biological Sciences, TIFR, Bangalore, India. [7]Center for Macroecology, Evolution and Climate, Globe Institute, University of Copenhagen, Copenhagen, Denmark. [8]Centre for Ecological Sciences, Indian Institute of Science, Bangalore, India. [9]Center for Evolutionary Hologenomics, Globe Institute, University of Copenhagen, Copenhagen, Denmark. [10]Center for Conservation Genomics, Smithsonian's National Zoo and Conservation Biology Institute, Washington, DC, USA. [11]Department of BioSciences, Rice University, Houston, TX, USA. [12]Novo Nordisk Foundation Center for Basic Metabolic Research, Faculty of Health and Medical Sciences, University of Copenhagen, Copenhagen, Denmark. [13]Illumina Laboratory Services, Illumina Inc., San Diego, CA, USA. [14]Carl R. Woese Institute for Genomic Biology, University of Illinois at Urbana-Champaign, Urbana, IL, USA. [15]Department of Ecology, Evolution, and Behavior, UIUC, Urbana, IL, USA. [16]Department of Anthropology, UIUC, Urbana, IL, USA. [17]Institute of Ecology and Evolution, University of Oregon, Eugene, OR, USA. [18]San Diego Zoo Wildlife Alliance, Escondido, CA, USA. [19]Department of Forestry, University of Dschang, Dschang, Cameroon. [20]Laboratoire de génétique de la faune, Agence nationale des parcs nationaux, Libreville, Gabon. [21]Save the Elephants, Nairobi, Kenya. [22]Department of Fish, Wildlife, and Conservation Biology, Colorado State University, Fort Collins, CO, USA. [23]Department of Biology, Queen's University, Kingston, ON, Canada. [24]Department of Environmental Management, Makerere University, Kampala, Uganda. [25]Department of Zoology, Entomology and Fisheries Sciences, Makerere University, Kampala, Uganda. [26]4718 SW Pendleton St, Portland, OR, USA. [27]Deceased: Iain Douglas-Hamilton. [28]These authors jointly supervised this work: Rasmus Heller, Hans Redlef Siegismund, Alfred L. Roca. ✉e-mail: pata.pecnerova@gmail.com

