## [Transparent Peer Review file · Nature Communications]

The genomic impact of population connectivity and decline in Africa's elephants

Corresponding Author: Dr Patrícia Pečnerová

Version 0:

Reviewer comments:

Reviewer #1

(Remarks to the Author)

Pečnerová et al. present a continent-scale demographic and landscape genomic analyses of African savannah and forest elephants across their native ranges. Using whole genome data from 232 elephants, they examine patterns of genetic divergence, gene flow, diversity, hybridization and demographic history of the two species.

This work is likely to be significant to field of conservation genetics, primarily as a high-quality dataset and statistical framework that will inform ongoing and future efforts to quantify the relative genetic health and relatedness of African elephant populations, as well as guide assisted migration and introduction efforts for both elephant species. The results of the study are thoroughly descriptive but not particularly surprising or novel in and of themselves. Mostly, they build on, clarify, and add a needed contextual framework to findings from previous studies by applying well established, and in some cases, updated statistical frameworks to a significantly larger, more geographically expansive, genome-scale dataset than has yet to be produced for the species.

The conclusions of the work are straight forward, and the statistical analyses applied toward those ends are both appropriate and rigorous. I found no glaring flaws in the data analyses presented. The interpretations and conclusion align well with the analytical results. The methodology meets the expected standards of the field and are explained clearly enough for the work to be reproduced by other genomics-oriented researchers. Taken together, the work is a thorough descriptive analysis conducted at an unprecedented scale, that will likely serve as the standard for basic population genetics questions for these species. The primary novelty of the work comes from the geographic scale of the genomic data, which will undoubtedly be an invaluable tool for future studies of the species.

Reviewer #2

(Remarks to the Author)

The manuscript entitled „The genomic impact of connectivity and decline in Africa's elephants” is a long-awaited and much-needed genome-scale update to the range-wide analysis of Modol et al. 2015. The data set is impressive, including more than 200 high coverage genomes. I congratulate the authors on their attempt to tie this complex, continent-wide data set together into a coherent manuscript.

Apart from their unusually large genome data set, the manuscript is reasonably written, the study is very interesting and the results important. Nature Comms. should be lucky to have this manuscript. I have only a few comments, some of which are major concerns, but I am certain that the authors will be able to address them adequately.

After spending all his time and money to obtain such a high quality genome alignment, why would that authors only produce an IBS tree? The IBS distance does not account for several molecular evolutionary processes, and more importantly, likelihood tools for tree reconstruction like IQTree and NGSdist have been available for years.

Also from Fig 2B, it is also clear that there are gradients in ancestry along the eastern part of Africa, with the eastern ancestry as far south as Kruger and along the northern savannas with western savanna ancestry as far east as northern Uganda and Eritrea. It appears that savanna elephants evolved into three main populations, with the southern population diverging further into three subpopulations. Unfortunately a tree of the relationships within each species is not provided and this

important information is unavailable to the reader (except if one looks at the base topology of the admixture graphs in the supplement, it certainly cannot be made out in Fig S26). These populations and subpopulations have had ongoing secondary contact with each other resulting in these gradient like patterns in ancestry. That also explains why the data look so IBD-like (Fig S15).

The approaches used to estimate migration events (qpGraph and Treemix) all rely on the assumption that the base tree (supplied or inferred) reflects common ancestry only. Can we be absolutely sure of that with African elephants? As far as I know, the latest elephantid phylogeny and evolutionary history is that of Polkapoulou et al. (2018). The most intriguing finding of that paper was the admixture between forest elephants and the now extinct *P. antiquus*. They determined the direction of gene flow (from forest to *P. antiquus*) using a similar graph approach as used in the present manuscript, but with a very limited number of genomes, so I always wondered whether that analysis would hold up with a larger data set. The PSMCs in that paper show increases in N_e in both *P. antiquus* and forest elephants. The same increase in forest N_e is confirmed by your larger data set (Fig. 7). How sure are the authors that there was no gene flow from *P. antiquus* back into forest elephants, resulting in the approx. 300kyr increase in forest N_e observed in Fig 7 and the much higher N_e of forest elephants (Fig 5)? Unfortunately, that scenario would mess up all the Dstats etc that were calculated in this manuscript, so it would be good to see some compelling evidence that the forest lineage did not experience immigration from a non-*Loxodonta* ancestor.

Section 362-386. Please also discuss how the different populations within each species are related to each other phylogenetically (point already made earlier in this review). That would also help figure out how the two species originally colonised their respective ranges.

Minor issues

Line 73. "The largest of them...." The forest elephant is smaller than the Asian elephant, and since you are calling them different species here, you will need to rephrase this sentence.

Line 95, "known-location" I think the correct term is "geo-referenced".

Line 139. Delete "scientific"

"While minuscule, the inferred forest ancestry in the genomes of savanna elephants was widespread, and appeared to align with geographical distance relative to the Congo-Guinean rainforest."

Did you actually test the above with a Mantel plot? It would be a nice addition to the manuscript and would provide more evidence to back up the authors interpretation of introgression patterns.

"In savanna elephants, the population structure appears to be consistent with isolation by distance (Fig. 4, Fig. S14, Fig. S15), a pattern also observed in pan-African genomic analyses of lions (Bertola et al. 2022), leopards (Pečnerová et al. 2021), and Cape buffalo (Quinn et al. 2023)."

I agree, perhaps the authors can think of a biological driver that links the mentioned species?

For forest elephants, there is a big sampling gap across the CAR and for savanna elephants, the main sampling gap is in south-central Africa including southern Tz, northern Mozambique, all of Zambia and Angola. Given the patterns of structure and connectivity, I don't think these sampling gaps are a major issue, but please add a sentence suggesting where animals sampled in these regions gap might fit into the structure as depicted in Fig 4a. Presumably they would simply fill in the gaps in the existing gradients.

Lines 527-534. This is an interesting hypothesis. *P. recki* is a species that dates from as early as the Pliocene, so this does not explain why savanna elephants underwent an increase in N_e while *P. recki* was around. Presumably there would also be an increase in N_e in more recent times as *P. recki* was becoming extinct? Perhaps this is one of several the reasons for the savanna elephants low N_e .

Line 872. It is not clear why one would want to investigate introgression in this particular direction. What informed this approach?

Why were forest elephants not also tested in positions H1 and H2? This might be obvious to the authors but not to the readers.

Line 877. The test is for introgression. So I am not sure why Garamba was left out, surely the Dstat for this test would be able to tell us which sampled savanna population could have been responsible for the Garamba introgression?

Section F4 ratios. Please explain better why you structured this test in this way. Is it because Lope and DzangaSanga are the least admixed forest populations? Is it because Sierra Leone and Lope/Dzanga are actually the sister taxa? I think this needs to be more clearly unpacked for the readers.

Please also be consistent with names, eg. the same samples are sometimes labelled Mozambique and sometimes Gorongoza. This all adds to confusion for the reader.

Line 893. I am intrigued by your definition of "Recently admixed". Wouldn't this expectation also hold for older (not recent) introgression events?

Figure 1. As far as I know Mashatu is in Botswana, not Zimbabwe.

Please rephrase the sentence starting in line 219, it took me a while to figure out that it was the between vs within species comparison.

One of the main downstream conservation applications for this data set would be to define African elephant conservation units. Only after the underlying evolutionary pattern of drift is estimated (that is, how many populations within each species and how are they related to each other), can connectivity and its extent be quantified and management decisions made. This ought to be mentioned in the last paragraph.

There are parts of the manuscript where the phrasing is awkward, and there seem to be more typos, grammatical errors and incorrect phrasing in the methods and supplemental sections compared to the rest of the MS, please address these before submitting a revision.

Version 1:

Reviewer comments:

Reviewer #2

(Remarks to the Author)

I have reviewed the resubmission of the manuscript „The genomic impact of connectivity and decline in Africa's elephants”.

I applaud the authors on the changes they have made, all of which satisfactorily answer my comments. I do not have any further major issues. I list below very minor points.

Lines 523- 525. It is certainly intriguing that both forest and savanna elephants show an increase in NE at this time despite their differences in preferred habitat. At any rate perhaps consider replacing the word “dominated” with “was present in large numbers”.

Line 532. “more data are needed”

Response to comment “Line 872”. In Fig S14, it is both interesting and unexpected that you find savanna introgression in DS and Lope. I am trying to figure out why this might be: ghost introgression, differential loss of alleles through drift, but the authors are right, it is premature to speculate without additional forest genomes.

Thanks also for including Garamba in Fig S14. The title of the bottom panel has forest spelled incorrectly.

REVIEWER COMMENTS

Reviewer #1 (Remarks to the Author):

Pečnerová et al. present a continent-scale demographic and landscape genomic analyses of African savanna and forest elephants across their native ranges. Using whole genome data from 232 elephants, they examine patterns of genetic divergence, gene flow, diversity, hybridization and demographic history of the two species.

This work is likely to be significant to field of conservation genetics, primarily as a high-quality dataset and statistical framework that will inform ongoing and future efforts to quantify the relative genetic health and relatedness of African elephant populations, as well as guide assisted migration and introduction efforts for both elephant species. The results of the study are thoroughly descriptive but not particularly surprising or novel in and of themselves. Mostly, they build on, clarify, and add a needed contextual framework to findings from previous studies by applying well established, and in some cases, updated statistical frameworks to a significantly larger, more geographically expansive, genome-scale dataset than has yet to be produced for the species.

The conclusions of the work are straight forward, and the statistical analyses applied toward those ends are both appropriate and rigorous. I found no glaring flaws in the data analyses presented. The interpretations and conclusion align well with the analytical results. The methodology meets the expected standards of the field and are explained clearly enough for the work to be reproduced by other genomics-oriented researchers. Taken together, the work is a thorough descriptive analysis conducted at an unprecedented scale, that will likely serve as the standard for basic population genetics questions for these species. The primary novelty of the work comes from the geographic scale of the genomic data, which will undoubtedly be an invaluable tool for future studies of the species.

>>> We thank the reviewer for the positive assessment of our work.

Reviewer #2 (Remarks to the Author):

The manuscript entitled „The genomic impact of connectivity and decline in Africa’s elephants” is a long-awaited and much-needed genome-scale update to the range-wide analysis of Modol et al. 2015. The data set is impressive, including more than 200 high coverage genomes. I congratulate the authors on their attempt to tie this complex, continent-wide data set together into a coherent manuscript.

Apart from their unusually large genome data set, the manuscript is reasonably written, the study is very interesting and the results important. Nature Comms. should be lucky to have this manuscript. I have only a few comments, some of which are major concerns, but I am certain that the authors will be able to address them adequately.

After spending all his time and money to obtain such a high quality genome alignment, why would that authors only produce an IBS tree? The IBS distance does not account for several molecular evolutionary processes, and more importantly, likelihood tools for tree reconstruction like IQTree and NGSDist have been available for years.

Also from Fig 2B, it is also clear that there are gradients in ancestry along the eastern part of Africa, with the eastern ancestry as far south as Kruger and along the northern savannas with western savanna ancestry as far east as northern Uganda and Eritrea. It appears that savanna elephants evolved into three main populations, with the southern population diverging further into three subpopulations. Unfortunately a tree of the relationships within each species is not provided and this important information is unavailable to the reader (except if one looks at the base topology of the admixture graphs in the supplement, it certainly cannot be made out in Fig S26). These populations and subpopulations have had ongoing secondary contact with each other resulting in these gradient like patterns in ancestry. That also explains why the data look so IBD-like (Fig S15).

Section 362-386. Please also discuss how the different populations within each species are related to each other phylogenetically (point already made earlier in this review). That would also help figure out how the two species originally colonised their respective ranges.

>>> We thank the reviewer for this suggestion. In the revised version of the manuscript, we have inferred maximum-likelihood phylogenetic trees based on autosomal SNPs using IQ-TREE. This includes a phylogenetic tree of both species, a savannah elephant-specific phylogenetic tree, and a forest elephant-specific phylogenetic tree.

In general, the phylogenetic relationships are influenced by many evolutionary processes, such as inter-specific and intra-specific gene flow, as well as incomplete lineage sorting. We observe that the gene flow detected in our study is influencing the topology of the trees, as best illustrated by the basal placement of hybrid populations of Garamba and the Queen Elizabeth National Park (Figs. S17, S18, S19). This makes inferring the divergence history complicated, as for example the forest ancestry in west-central savanna elephants is influencing the inferred phylogeny (Fig. S18). Therefore, we prefer not to draw major conclusions about the origin of the clades.

We have included a paragraph in the main text (lines 292-297) which reads as follows: "A potentially complex pattern of bidirectional gene flow between species was

corroborated by admixture graph analyses in qpGraph in ADMIXTOOLS2 (Fig. S15) and Treemix (Fig. S16), and this makes inferences regarding the phylogeny of African elephants challenging, as illustrated by the basal placement of the hybrid populations of Garamba and the Queen Elizabeth National Park in the autosomal phylogenetic tree inferred using IQ-TREE (Fig. S17, Fig. S18, Fig. S19), which is most likely driven by the high proportion of mixed ancestry.”

The approaches used to estimate migration events (qpGraph and Treemix) all rely on the assumption that the base tree (supplied or inferred) reflects common ancestry only. Can we be absolutely sure of that with African elephants? As far as I know, the latest elephantid phylogeny and evolutionary history is that of Polkapoulou et al. (2018). The most intriguing finding of that paper was the admixture between forest elephants and the now extinct *P. antiquus*. They determined the direction of gene flow (from forest to *P. antiquus*) using a similar graph approach as used in the present manuscript, but with a very limited number of genomes, so I always wondered whether that analysis would hold up with a larger data set. The PSMCs in that paper show increases in N_e in both *P. antiquus* and forest elephants. The same increase in forest N_e is confirmed by your larger data set (Fig. 7). How sure are the authors that there was no gene flow from *P. antiquus* back into forest elephants, resulting in the approx. 300kyr increase in forest N_e observed in Fig 7 and the much higher N_e of forest elephants (Fig 5)? Unfortunately, that scenario would mess up all the Dstats etc that were calculated in this manuscript, so it would be good to see some compelling evidence that the forest lineage did not experience immigration from a non-Loxodonta ancestor.

*>>> Thank you for this interesting suggestion. Unfortunately, even though we have sampled many more genomes compared to Palkopoulou et al., in terms of represented forest populations our study is not a significant expansion of the available data. The two forest genomes in Palkopoulou et al. originate from Sierra Leone and Dzanga Sangha, two out of five forest populations also represented in our study. Moreover, Palkopoulou et al. showed that the gene flow with *Palaeoloxodon* relates specifically to the west African forest elephant from Sierra Leone, and that is also the same genome that we are using as the only representative of west Africa in our dataset. Thus, we could only attempt to test for gene flow between *Palaeoloxodon* and Central forest elephants of Lope, Odzala, and Garamba (the three additional forest populations in our study), but we consider it unlikely that we would discover a signal that the Palkopoulou study failed to identify in Dzanga Sangha. Moreover, such a test would be mostly validating the conclusions of Palkopoulou et al. that the gene flow with *Palaeoloxodon* was asymmetric towards the west African forest elephants, which we consider beyond the scope of the current study.*

*As the reviewer suggests, it would be beneficial to identify the directionality of the gene flow and whether any *Palaeoloxodon* ancestry might be present in forest elephants. However, similarly as in Palkopoulou et al., performing D-statistics with *Palaeoloxodon* would merely tell us if they share alleles, but would not advise on the direction of the gene flow. To infer that, we would have to perform elaborate admixture graph analyses but it is uncertain that we would be able to better infer the proportions and directionality of ancestry components, which were already extensively tested in the Palkopoulou study, and to do so would require extensive analyses including other proboscideans, which again, we think is beyond the scope of our already broad study.*

To express the potential influence of unaccounted admixture events on the PSMC results, we modified the text (lines 512-517) to the following: “It is important to take into consideration that the population trajectories might be influenced by unaccounted admixture events, as such events are not uncommon in proboscideans while the timing of the population trajectories can be influenced by imprecise estimates of generation time and mutation rate. Even so, our results suggest that the split between the savanna and the forest elephant occurred around 4 million years ago (Mya), falling into the inferred split time of $\approx 2-5$ Mya estimated by Palkopoulou et al..”

*Regarding the effect of gene flow from *P. antiquus* into forest elephants on our Dstats, while we agree that this would be a case of “ghost introgression” that could have an effect on the absolute value of the Dstats, we do not think it could invalidate our main conclusion of widespread gene flow from forest to savanna elephant populations that diminishes with distance to the forest. This conclusion is based on Dstat with the shape $((S1, S2), F), A$, where S1 and S2 stand for different savanna individuals, F for a forest individual and A an asian elephant. Changes in the ancestry of F might have an impact on the absolute values of the D-statistics, but as long as that gene flow does not affect the savanna populations, it would not have an impact on relative comparisons of the values when we have S1 and S2 from different populations, which is the signal our conclusion is based on.*

Minor issues

Line 73. “The largest of them....” The forest elephant is smaller than the Asian elephant, and since you are calling them different species here, you will need to rephrase this sentence.

>>> Thank you for spotting this error. We have rephrased the first paragraph to following: “Megaherbivores are especially vulnerable to extinction, and the loss of large land mammals can have cascading effects on entire ecosystems. The largest of them all, elephants, are among the most iconic species threatened by extinction in the Anthropocene. Elephants in Africa are now recognized as two species, the African

savanna elephant (Loxodonta africana) and the African forest elephant (L. cyclotis)."

Line 95, "known-location" I think the correct term is "geo-referenced".

>>> *The term was corrected to "geo-referenced".*

Line 139. Delete "scientific"

>>> *We deleted "scientific".*

"While minuscule, the inferred forest ancestry in the genomes of savanna elephants was widespread, and appeared to align with geographical distance relative to the Congo-Guinean rainforest."

Did you actually test the above with a Mantel plot? It would be a nice addition to the manuscript and would provide more evidence to back up the authors interpretation of introgression patterns.

>>> *Thank you for this suggestion. We have now added a Mantel test and we included the following text (lines 257-262) in the manuscript: "While minuscule, the inferred forest elephant ancestry in the genomes of savanna elephants was widespread geographically, and a Mantel test revealed a weak but significant correlation between forest ancestry and geographic distance to the Congo-Guinean forest (Spearman's $\rho = 0.234$, $p = 1e-04$; Fig. S13), indicating that forest elephant-associated ancestry tends to decline with increasing distance from the tropical forest." And we describe the approach in the Methods: "We tested the correlation between F4 ancestry and geographic distance-to-tropical forest. We used the WWF Terrestrial Ecoregions of the World shapefile to extract and merge polygons corresponding to Congo-Guinean rainforests, and then we calculated great-circle distances to the nearest rainforest boundary. Pairwise Euclidean distance matrices were then computed for (i) ancestry proportions and (ii) distance-to-rainforest values, with an additional matrix of straight-line geographic distances between sampling sites. Finally we applied the Mantel test (with Spearman correlation and 9,999 permutations) in R v4.4.3 using the package vegan v2.7-2 in Rstudio v2024.12."*

"In savanna elephants, the population structure appears to be consistent with isolation by distance (Fig. 4, Fig. S14, Fig. S15), a pattern also observed in pan-African genomic analyses of lions (Bertola et al. 2022), leopards (Pečnerová et al. 2021), and Cape buffalo (Quinn et al. 2023)."

I agree, perhaps the authors can think of a biological driver that links the mentioned species?

>>> Thanks for this suggestion. We included a hypothesis on the drivers of this pattern (line 344-346): “This pattern is consistent with the long-distance dispersal of these species, in which philopatric females maintain related social groups, while dispersing males move alleles across the landscape, resulting in a gradient of genetic differentiation with distance.”

For forest elephants, there is a big sampling gap across the CAR and for savanna elephants, the main sampling gap is in south-central Africa including southern Tz, northern Mozambique, all of Zambia and Angola. Given the patterns of structure and connectivity, I don't think these sampling gaps are a major issue, but please add a sentence suggesting where animals sampled in these regions gap might fit into the structure as depicted in Fig 4a. Presumably they would simply fill in the gaps in the existing gradients.

>>> While we prefer not to speculate on the particular placement of these locations, we agree that these will likely contribute to further “smoothen” the isolation by distance pattern, and thus, we added the following statement (line 372-374): “We hypothesize that future sampling from locations, which are missing in our dataset, will further contribute to the continuity of the isolation by distance pattern of genetic variation.”

Lines 527-534. This is an interesting hypothesis. *P. recki* is a species that dates from as early as the Pliocene, so this does not explain why savanna elephants underwent an increase in N_e while *P. recki* was around. Presumably there would also be an increase in N_e in more recent times as *P. recki* was becoming extinct? Perhaps this is one of several the reasons for the savanna elephants low N_e .

*>>> We agree that this part is rather speculative, but we tried to address this hypothesis, as it was previously proposed to explain the observed pattern and remains one of the leading hypotheses. One explanation for why we see an increase in N_e in the savanna elephant around 4-3 Mya despite *P. recki* already being present, is that at the time, the open environment has not yet reached its more extreme condition, and perhaps the relatively more wooded open environment during the time would have favoured savanna elephants. Another potential explanation is that *Palaeoloxodon* dominated in East Africa, but perhaps savanna elephants at the time were more abundant in other parts of the continent, even in the forested regions. As to whether we would see an increase in recent times, that is rather uncertain due to the PSMC's notorious lack of robustness in reconstructing recent events, and according to some, *Palaeoloxodon* survived late Pleistocene. Thus, we would prefer to refrain from interpreting the PSMC curve in the most recent time periods, and are very reluctant to use it as an argument regarding *P. recki*'s extinction.*

*We tried to clarify this in the text in the following way (lines 519-533): “Savanna elephant effective population size peaked around 3 Mya, which coincides with the African climate becoming periodically cooler and drier and arid-adapted biota spreading 2.8 Mya. Interestingly, *P. recki*—a more specialized grazer—is thought to have dominated the open environments of East Africa at that time. Thus, it remains unclear if African elephants reached high population sizes due to their dietary flexibility during the onset of open environments or dominated in other parts of the continent.*

*The effective population size of savanna elephants declined drastically after 2 Mya and remained low ever since (Fig. 7). We speculate that when the extent of open grasslands reached its peak \approx 1.8 Mya, *P. recki* outcompeted the savanna elephant and pushed it to refugia until its extinction towards the end of Pleistocene. It has been hypothesized that it was the more generalized diet of mixed-feeder/browser savanna elephant that allowed it to persist after the extinction of *P. recki*. However, we caution that while our results roughly align with the palaeontological record, more data is needed to clarify whether other factors contributed to the low effective population sizes of savanna elephants.”*

Line 872. It is not clear why one would want to investigate introgression in this particular direction. What informed this approach?

Why were forest elephants not also tested in positions H1 and H2? This might be obvious to the authors but not to the readers.

>>> The choice to investigate forest ancestry in savanna elephants using D-statistics was driven by detection of low proportions of forest ancestry in savanna elephants in Serengeti and Zambezi in the Admixture analysis, which was further confirmed in the D-statistics. There was no indication of widespread admixture in the opposite direction, therefore we did not test it with D-statistics. This was clarified in the text (lines 222-224): “Prompted by the observation of forest ancestry in some individuals in Serengeti and Zambezi in the Admixture analyses (Fig. 2, Table S4), we tested whether any savanna elephant population showed an excess of allele sharing with the forest elephant.”

However, we agree that this should be formally tested and we have now performed D-statistics analyses of excessive savanna alleles in forest elephant populations, finding that two populations outside of the hybrid zone, Dzanga Sangha and Odzala, show evidence of savanna admixture. We added the D-statistics as Fig. S14 and we report this finding in the text (lines 282-291): “We also tested whether any forest elephant population showed an excess of alleles shared with the savanna elephant. We tested all combinations of forest elephants in H1 and H2, and placed savanna elephants in H3 as a relatively non-admixed savanna elephant population (Kruger). As expected, we detected a signal of savanna elephant ancestry in the hybrid population of Garamba; however, we also detected signals of savanna ancestry in Dzanga Sangha and Odzala

(Fig. S14), suggesting that the gene flow is bidirectional even outside of the hybrid zone, although this was not evident in PCA or Admixture analyses (Fig. 2). However, the coarse representation of forest elephants in our dataset, and the lack of savanna elephant mtDNA introgression in forest elephants²³, prevents us from drawing conclusions about the broader patterns of savanna introgression in forest elephants.”

Line 877. The test is for introgression. So I am not sure why Garamba was left out, surely the Dstat for this test would be able to tell us which sampled savanna population could have been responsible for the Garamba introgression?

>>> In the context of studying forest to savanna introgression, including Garamba will create a confounding effect since it carries large amount of ancestry from a savanna population, and it would not be possible to separate allele sharing caused by the savanna ancestry in Garamba from allele sharing caused by introgression from other forest populations to savanna. For a similar reason D-statistics would also not be able to tell what population is the source of the savanna admixture in Garamba, since all potential donors (savanna populations close to forest elephant range) carry some forest elephant ancestry, and it would not be possible to distinguish if an increase in D-statistics is due to highest forest ancestry in the savanna population, or the savanna population being genetically closer to the savanna population that introgressed into Garamba. To do that, we would need to infer local ancestry and analyse the different types of ancestry tracts separately, which is outside the scope of this manuscript.

Section F4 ratios. Please explain better why you structured this test in this way. Is it because Lope and DzangaSanga are the least admixed forest populations? Is it because Sierra Leone and Lope/Dzanga are actually the sister taxa? I think this needs to be more clearly unpacked for the readers.

>>> Thank you for pointing out the lack of clarity. We rewrote the description (lines 883-891) to specify: “The F4 ratio test estimates admixture proportion (α) in a five-population scenario as $\alpha = f_4(A, O; X, B) \div f_4(A, O; C, B)$, including a test population (X; all savanna populations), an outgroup (O; Asian elephant), a reference “unadmixed” savanna population (B; Kruger), a reference “unadmixed” forest population 1 (A; Sierra Leone as west forest population), and a reference “unadmixed” forest population 2 (C; Lope combined with Dzanga Sangha as central forest population). In other words, we estimated F4 ratios in the form of $\alpha = f_4(\text{Sierra Leone, Asian; target, Kruger}) / f_4(\text{Sierra Leone, Asian; Lope/Dzanga Sangha, Kruger})$. We used 5×10^6 as the SNP block size for jackknifing.”

Please also be consistent with names, eg. the same samples are sometimes labelled Mozambique and sometimes Gorongoza. This all adds to confusion for the reader.

>>> *We agree that it is important to maintain consistency. While sometimes it is necessary to switch between the geographic context of individual locations and broader regions like countries, we tried to clarify and simplify these in the text. For example, for the Gorongosa National Park in Mozambique, we did the following corrections: 1) replaced “Gorongosa” with “Mozambique” as we consistently use “Mozambique” in the figures, “The locations in southern Africa formed distinctive southeastern (MozambiqueGorongosa, Kruger, Mashatu),...”. 2) We included country names in the following sentence: “...in southeastern Africa, the Gorongosa National Park (Mozambique), Mashatu Game Reserve (Botswana), and Kruger National Park (South Africa) form...”*

Line 893. I am intrigued by your definition of “Recently admixed”. Wouldn’t this expectation also hold for older (not recent) introgression events?

>>> *In the context of the method used (apoh), recent admixture is defined as the two parents of the individuals having different admixture proportions. In older admixture events, interbreeding, recombinations and potentially selection will have broken up the admixture tracts and mixed it so the parents will have the same admixture proportions. There are cases (for example the offspring of two F1 hybrids, and in general any cases of the mating between two recent hybrids carrying the same admixture proportions) where recent admixture would not be detected by that method, however for older introgression we always expect that the ancestry is mixed up and/or diluted enough to not have differences in the admixture proportions of the parents.*

Figure 1. As far as I know Mashatu is in Botswana, not Zimbabwe.

>>> *Thank you for pointing out this mistake, the figure was corrected.*

Please rephrase the sentence starting in line 219, it took me a while to figure out that it was the between vs within species comparison.

>>> *We clarified the calculation by modifying the statement (now on lines 202-206) to: Finally, the FST between species, calculated as the average pairwise FST between locations of the two species, was 0.64. By contrast, the within-species FST, calculated as the average of all pairwise comparisons between locations within each species, was much smaller: 0.067 in savanna elephants and 0.054 in forest elephants (Fig. S5).”*

One of the main downstream conservation applications for this data set would be to define African elephant conservation units. Only after the underlying evolutionary pattern of drift is estimated (that is, how many populations within each species and how are they related to each other), can connectivity and its extent be quantified and management decisions made. This ought to be mentioned in the last paragraph.

>>> Thank you for this suggestion. We included it in the paragraph (lines 604-606) as follows: “Second, by describing the genetic composition of each species, this dataset lays the groundwork for defining conservation units as a prerequisite of future conservation actions.”

There are parts of the manuscript where the phrasing is awkward, and there seem to be more typos, grammatical errors and incorrect phrasing in the methods and supplemental sections compared to the rest of the MS, please address these before submitting a revision.

>>> Thank you for pointing this out, we performed a thorough proof-checking and language editing to the best of our ability.

REVIEWERS' COMMENTS

Reviewer #2 (Remarks to the Author):

I have reviewed the resubmission of the manuscript „The genomic impact of connectivity and decline in Africa’s elephants”.

I applaud the authors on the changes they have made, all of which satisfactorily answer my comments. I do not have any further major issues. I list below very minor points.

>>> *We kindly thank the reviewer for their positive assessment.*

Lines 523- 525. It is certainly intriguing that both forest and savanna elephants show an increase in NE at this time despite their differences in preferred habitat. At any rate perhaps consider replacing the word “dominated” with “was present in large numbers”.

>>> *The sentence has been corrected accordingly and now reads: “Interestingly, *P. recki*—a more specialized grazer—is thought to have been present in large numbers in the open environments of East Africa at that time.”*

Line 532. “more data are needed”

>>> *Corrected.*

Response to comment “Line 872”. In Fig S14, it is both interesting and unexpected that you find savanna introgression in DS and Lope. I am trying to figure out why this might be: ghost introgression, differential loss of alleles through drift, but the authors are right, it is premature to speculate without additional forest genomes.

>>> *We agree. While it is tempting, we believe more data is needed to address this with confidence.*

Thanks also for including Garamba in Fig S14. The title of the bottom panel has forest spelled incorrectly.

>>> *Thanks for pointing this out, we corrected the figure.*